# Learning syntax without semantics: Disentangled tiny language models

**Ezra Winston** [1]  **J. Zico Kolter** [1]

## Abstract

Language models acquire syntax and world knowledge together, entangling the two in ways that limit efficiency and controllability. We show that syntax can be learned while suppressing semantic plausibility and world-knowledge cues, yielding more efficient and controllable models. We train tiny LMs on grammatical nonsense — syntactically well-formed text with semantic content ablated via constrained relexicalization (SAMBAL). Models trained on this data perform comparably to standard pretraining on syntactic benchmarks (BLiMP, SyntaxGym) while scoring at chance on world knowledge probes (EWoK). On targeted grammar-plausibility conflict probes, content-neutral models prefer grammaticality where standard models prefer plausibility, and their representations show more syntactic vs. lexical alignment. On efficiency, disentanglement yields substantial sample and parameter gains in low-resource regimes: a 5M-parameter model matches a 30M-parameter baseline at the same data budget. On controllability, content-neutral models adapt rapidly to a new domain with minimal exposure, suggesting the feasibility of modular post-hoc knowledge specialization.

## 1. Introduction

Humans acquire core language abilities from vastly less linguistic exposure—often estimated at $\lesssim 10^8$ words—than modern language models, which are typically trained on $10^{11}$–$10^{12}+$ tokens (Charpentier et al., 2025b; Hu et al., 2024a; Hoffmann et al., 2022). One key difference is that LMs are trained to serve multiple roles, as statistical models of linguistic form, advanced reasoners, and repositories of world knowledge and semantic plausibility (Mahowald et al., 2024; Bender & Koller, 2020). Whether syntax can

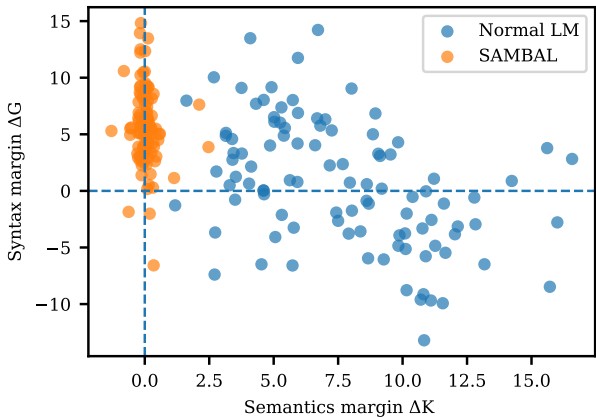

*Figure 1.* **Syntax margin vs. semantic-plausibility margin on minimal sentence pairs.** Models trained on grammatical nonsense (SAMBAL, orange) show a preference for syntactically-correct sentences, with positive logit margin $\Delta G = \log P\,(\text{grammatical}) - \log P\,(\text{ungrammatical})$, but no preference for semantically-plausible sentences, with logit margin $\Delta K = \log P\,(\text{plausible}) - \log P\,(\text{implausible})$ near zero, unlike a normally trained LM (§4.3).

be learned independently of meaning is a central question in linguistics, spanning arguments for the autonomy of syntax, poverty-of-the-stimulus debates, and evidence that learners can extract structure from distributional regularities (Chomsky, 1957; 1965; Pullum & Scholz, 2002; Saffran et al., 1996). This raises a concrete question for language modeling: can syntax be learned more efficiently if we actively suppress semantic and world-knowledge cues?

If syntax can be isolated from knowledge, two practical benefits follow. First, *efficiency*: if syntax requires less data than syntax-plus-knowledge, then isolating syntax learning it could yield sample and parameter gains—relevant both for low-resource settings and for understanding what inductive biases support language learning (Kaplan et al., 2020; Hoffmann et al., 2022; Warstadt et al., 2023; Hu et al., 2024a; Charpentier et al., 2025b). Second, *controllability*: knowledge learned parametrically is notoriously hard to update, edit, or specialize for new domains (Meng et al., 2022; 2023). Prior work addresses these problems post hoc—via steering, editing, and unlearning (Liu et al., 2024; Zhao et al., 2024; Wang et al., 2024; Yao et al., 2024). Our work suggests a

---

[1]Machine Learning Department, Carnegie Melon University, Pittsburgh, PA, USA. Correspondence to: Ezra Winston <ewinston@cs.cmu.edu>.

*Proceedings of the 43rd International Conference on Machine Learning*, Seoul, South Korea. PMLR 306, 2026. Copyright 2026 by the author(s).

different approach: suppress knowledge acquisition during pretraining, yielding a syntax-competent base onto which domain knowledge can later be added modularly.

**Our approach.** We hypothesize that a tiny LM can learn *syntactic competence* while suppressing *semantic plausibility and world-knowledge cues*. To test this, we train on *grammatical nonsense*—text that is syntactically well-formed yet semantically anomalous, in the spirit of Chomsky's classic "colorless green ideas" (Chomsky, 1957; Gulordava et al., 2018). Our pipeline, SAMBAL[1], transforms natural text into grammatical nonsense through constrained relexicalization: SAMBAL preserves morphosyntax, agreement, and complement structure (c-selection) while ablating plausibility priors, factual associations, and world-knowledge cues (s-selection). For example:

> **Original:** *Adopting instead of shopping reduces demand for commercially bred pets and gives loving homes to those already waiting.*
>
> **SAMBAL:** *Eating instead of learning gives probability for occasionally balanced men and creates architectural tears to those now living.*

We pretrain tiny LMs on SAMBAL-transformed corpora of 10M tokens or less, and evaluate them along several axes.

**Contributions and findings. (1) Syntactic competence.** SAMBAL matches standard pretraining on the grammar benchmarks BLiMP (Warstadt et al., 2020) (79.3% vs. 79.2% acc.) and most SyntaxGym (Gauthier et al., 2020) suites (79.8% vs. 78.6% acc.), with the notable exception of reflexive SyntaxGym suites. **(2) Semantic suppression.** SAMBAL falls to chance on EWoK (Ivanova et al., 2025) (50.8% vs. 54.3%) and shows near-chance plausibility preference on swap probes (55% vs. 100%). **(3) Reduced entanglement.** On grammar–plausibility conflict probes, SAMBAL prefers grammaticality where the baseline model prefers plausibility; representations show higher alignment to syntax relative to semantics. **(4) Efficiency gains.** SAMBAL is more efficient across a range of very small data and parameter budgets; with 1M words, a 5M-parameter SAMBAL model matches a 30M-parameter baseline. **(5) Modular specialization.** SAMBAL models (by design) have high perplexity on ordinary text, but they adapt rapidly via LoRA (Hu et al., 2022) to small domain-specific corpora—both 580k words of *Lord of the Rings* (Tolkien, 1994) and a similarly-sized set of PubMed abstracts (NLM, 2024)—a regime where training a competent model from scratch would be infeasible.

---

[1] https://github.com/ezrawinston/sambal

## 2. Background and related work

Our approach is motivated by the hypothesis that syntax and world knowledge, while entangled in standard pretraining, may be realized as distinct mechanisms that can largely be teased apart. This connects with several established research threads: (i) the form–meaning distinction in language, (ii) separability of syntax in model representations, (iii) child-scale / tiny-LM efficiency, (iv) shaping the pretraining signal toward structure, and (v) modular, updatable knowledge.

**Syntax, semantics, and separability.** The distinction between linguistic form and communicative meaning is a foundational debate in cognitive science. Classically, the *autonomy of syntax* hypothesis posits that the combinatorial rules of language operate independently of concepts or communicative intent (Chomsky, 1957; 1965). This is famously illustrated by the sentence "Colorless green ideas sleep furiously," which speakers recognize as grammatically well-formed despite its semantic vacuity. This separation of form from meaning has been argued to persist in language models, which can master linguistic form without correspondingly grounded meaning (Bender & Koller, 2020). Relatedly, Mahowald et al. (2024) distinguish *formal linguistic competence*—knowledge of linguistic rules and patterns—from *functional linguistic competence*—non-linguistic cognitive abilities such as formal reasoning and world knowledge.

**Separability of syntax in model representations.** Whether trained models represent syntax independently of meaning has been studied largely through their internal representations. Structural probes recover syntactic dependency trees linearly from contextual embeddings, and this signal appears genuinely structural rather than a reflection of semantic predictability (Hewitt & Manning, 2019; Diego-Simón et al., 2025). Consistent with separability, syntactic structure remains decodable even when lexical meaning is removed: under grammaticality-controlled nonce (*Jabberwocky*) relexicalization, decoding drops only modestly and most of the syntactic signal is preserved (Arps et al., 2024), though earlier pseudoword studies reported larger drops (Maudslay & Cotterell, 2021). Grammaticality is itself a concern here: relexicalizing on part-of-speech alone frequently breaks argument structure, yielding sentences rated grammatical only ∼38% of the time (Gulordava et al., 2018; Arps et al., 2024), which motivates the careful design of our pipeline (Appendix A). Overall, these studies probe whether syntax is *decodable* from a fixed model after meaning is removed at evaluation time; we instead remove meaning throughout *training* and ask whether syntax is still *learnable*.

**Tiny LMs and child-scale pretraining.** Whether syntax can be acquired from the limited input scale available to

children—or instead requires innate structural bias—is long debated: poverty-of-the-stimulus arguments hold that the input underdetermines the grammar (Pullum & Scholz, 2002), whereas infants demonstrably extract structural regularities from distributional cues alone (Saffran et al., 1996). Language models at comparable data scales ($< 100M$ words) are the focus of efforts such as the *BabyLM Challenge* (Warstadt et al., 2023; Hu et al., 2024b). Models with 10M–100M parameters can achieve strong BLiMP performance, especially with sample-efficient objectives (e.g., hybrid causal/masked objectives) and strong optimization or distillation (Warstadt et al., 2023; Charpentier & Samuel, 2024; Hu et al., 2024b; Timiryasov & Tastet, 2023). These findings have inspired *Fundamental Language Models* (FLMs)—compact models trained for linguistic proficiency while minimizing stored world knowledge (Collado-Montañez et al., 2025). We build on these architectures and objectives (GPT-BERT), while explicitly enforcing suppression of world knowledge during training.

**Shaping the pretraining signal for syntax.** A complementary line of work improves grammatical learning by shaping the pretraining *signal* or objective rather than the model, through explicit structural supervision or auxiliary syntactic objectives (Wilcox et al., 2020; Lindemann et al., 2024). Most relevant to us are interventions on the training *data* itself: pretraining on child-directed speech imparts a stronger hierarchical-syntax inductive bias than standard corpora (Mueller & Linzen, 2023), and pre-pretraining on a formal language whose hierarchical structure mirrors natural language transfers a structural bias that improves syntactic generalization and natural-language sample-efficiency (Hu et al., 2025). These approaches motivate our data-centric view but pursue a different aim: where they add structural bias to improve grammar acquisition, SAMBAL removes semantic content to isolate syntax from meaning.

**Controllability and modular knowledge.** A growing literature aims to make model behavior easier to update and specialize without rewriting a base model. Parameter-efficient adaptation (e.g., LoRA-style adapters) enables low-cost, composable specialization (Hu et al., 2022), while post-hoc interventions such as steering, model editing, and unlearning can modify or remove targeted behaviors after pretraining (Meng et al., 2023; Zhao et al., 2024). Our approach is more direct, ablating the pretraining data so plausibility and world knowledge are largely absent during pretraining. Our small-domain adaptation studies on *Lord of the Rings* and PubMed are a small step toward modularity: a content-neutral syntactic base should admit rapid, low-data domain adaptation by leveraging its syntactic competence.

*Table 1.* What SAMBAL preserves vs. ablates. Borderline items are meaning-adjacent but are retained because they directly affect acceptability.

| |
|---|
| **Preserve:** Closed-class items; NPIs and NPI licensors; protected MWEs; morphosyntactic features; agreement; complement types / subcategorization (c-selection); raising/control/ECM signatures |
| **Borderline:** Grammaticized features that look semantic but affect well-formedness (e.g., gender and humanness for agreement/binding; noun mass/count class when determiners depend on them) |
| **Ablate:** Named entities; topical content; selectional preferences (s-selection); factual/encyclopedic associations; typical event/role co-occurrence patterns |

## 3. SAMBAL: grammatical nonsense via constrained relexicalization

### 3.1. Goals and terminology

SAMBAL targets *grammatical nonsense*: outputs that remain syntactically well-formed while suppressing cues that support world knowledge or semantic plausibility. By *syntax*, we mean the formal constraints that govern well-formedness (categories/features, dependencies, agreement, binding, complement structure); by *semantics* we mean the meaning-bearing content (open-class lexical meaning, compositional interpretation, and background knowledge that makes an utterance informative or plausible).[2]

**What we preserve vs. ablate.** The distinction of which aspects of language our pipeline should preserve or ablate is nuanced, and we defer details to Appendix K and summarize our default partition in Table 1. We first preserve core morphosyntactic and licensing constraints that maintain grammatical well-formedness (e.g., agreement morphology: *this dog runs* vs. *this dog run*; closed-class items such as determiners/auxiliaries/complementizers (e.g. *a, the, has, are, that, whether*); and binding/licensing configurations such as reflexives and Negative polarity items (NPIs) that are typically only acceptable in certain "licensing" environments (*I didn't see any dogs* vs. *I saw any dogs*).

When relexicalizing open-class items (content nouns/verbs/adjectives/adverbs), our core operational principle is: *preserve c-selection, ablate s-selection*. By *c-selection* we mean lexically governed requirements on complement *category* and realization (e.g., whether a

---

[2]We often use *grammar* and *syntax* interchangeably. Also, we use *semantics* in the colloquial sense, excluding the structural, compositional phenomena (e.g. quantifier scope, negation, binding) that formal semantics treats and that SAMBAL in fact preserves (see Appendix K and §4.3).

**Algorithm 1** Constrained relexicalization (SAMBAL) for a sentence $x = [x_1, \ldots, x_n]$

---

1: Parse sentence $x$ to obtain tokens $[t_1, \ldots, t_n]$, POS/morph, dependencies.
2: Freeze protected tokens/spans (e.g. closed class and MWEs).
3: **for** each replaceable token $t_i$ **do**
4:   Compute syntactic context key $k(t_i)$ (POS, morph, dep role, head features, frame features).
5:   Retrieve candidate lemma set $C(k(t_i))$ from bucketed lexicon (and optional to-inf/control filters).
6:   Optionally intersect $C_i = C(k(t_i))$ with context-conditioned candidate set from corpus stats.
7:   Sample replacement lemma $\ell_i \sim \text{Sampler}(C_i)$.
8:   Realize surface form $s(\ell_i)$ via morphological inflection under original morph features.
9: **end for**
10: Optionally, round-trip parse and reject-resample up to $N$ times if invariants fail.
11: Output augmented sentence $x' = [s(\ell_1), \ldots, s(\ell_n)]$.

---

verb takes an NP/PP/CP/to-infinitive: *depend on [PP]*, *believe [CP]*, *want [to VP]*; and raising/control/ECM signatures that affect argument realization, e.g., *seem [to VP]* vs. *persuade NP [to VP]* vs. *believe NP [to VP]*). By *s-selection* we mean constraints and preferences on the semantic type of arguments and events (selectional restrictions and plausibility, e.g., *devour* typically takes an edible object; *repair* an artifact), which are intentionally suppressed.

**Borderline cases.** Some distinctions (notably gender/humanness/animacy, and polarity sensitivity via NPIs/licensors) are meaning-adjacent yet tightly coupled to syntactic acceptability. We therefore preserve them by default; Appendix K motivates this choice and gives concrete examples.

### 3.2. Constrained relexicalization pipeline

SAMBAL performs constrained relexicalization by replacing eligible open-class tokens with new lemmas sampled from constraint-matched pools, then realizes surface forms to match the original morphosyntactic features (see Algorithm 1). We highlight a few key aspects here and defer details to Appendix A.

**Replacing from syntactically-constrained candidate sets.** For each replaceable token, we compute a *syntactic context key* from the parse (POS, morphosyntactic features, dependency role, and—when available— subcategorization cues) and retrieve an initial candidate lemma set from a *bucketed lexicon* built from external resources (e.g. WordNet (Miller,

1995)/VerbNet (Schuler, 2005)). These resources let us restrict replacements to lemmas with compatible syntactic behavior, preserving *c-selection* (e.g., sampling PP-selecting verbs for *depend on [PP]*, CP-taking verbs for *believe [CP]*, and *to*-infinitival verbs for *want [to VP]*), while leaving semantic fit unconstrained.

In practice, lexicon coverage can be sparse and idiosyncratic, so recorded frames may reflect rare or lexicalized uses rather than typical distribution—for example, a verb like *smile* is almost always intransitive, yet can appear with particular object-like complements (*smiled a smile* vs. *smiled a book*). Moreover, for certain constructions we wish to control (e.g., inventories of NPI licensors, or fine-grained *to*-infinitival buckets for *to*-infinitival verbs/adjectives), existing lexicons are unavailable or incomplete; in these cases we construct targeted lexica and bucket definitions ourselves (Appendix A).

**Context-conditioned sampling.** A key challenge is that naïve swapping from external lexicons can still yield frequent ungrammatical outputs (e.g., *smiled a book*) even when we impose strong syntactic constraints. In practice there is a tradeoff: if we make lexicon/frame matching *permissive* to ensure coverage admits candidates whose recorded frames are noisy or idiosyncratic and thus produce ill-formed sentences; if we make matching *strict*, matching means many contexts have *no* lexicon-attested candidates at all. We mitigate this by context-conditioned sampling with a *context key* (POS/morph/dependency/frame features) using corpus statistics from our original training corpus. A context-key *back-off chain* (from fine to coarse feature bundles) is defined, and we back off until the candidate pool exceeds a minimum size, then sample replacement lemmas in proportion to their observed frequency in that backed-off context.

**Round-trip validation and reject–resample.** Finally, we optionally reparse the augmented sentence and reject/resample if key invariants fail (e.g., POS/morph mismatches, or violations of protected configurations such as NPI licensing). This lightweight round-trip check acts as an additional guardrail against noisy parsing decisions and occasional realization errors introduced by relexicalization, without requiring manual rules for every construction.

## 4. Experiments

### 4.1. Setup

All of our experiments are based on the 10M-word `baby-cosmo-fine` corpus and small GPT-BERT hybrid model (30M params), the winning combination from the 2024 BabyLM competition (Charpentier et al., 2025a; Hu et al., 2024a). GPT-BERT (Charpentier & Samuel, 2024) is

a hybrid causal/masked language model that combines GPT-style autoregressive and BERT-style masked objectives in a single transformer stack, which is state-of-the-art on BLiMP at this scale. Baseline models are trained on the original corpus, and SAMBAL models are trained identically, but on the corpus that has been fully relexicalized according to Algorithm 1. Default SAMBAL settings are reported in Appendix A and ablations of various pipeline components are described in §5 and Appendix C.

We follow prior small-data baselines by reporting both a **short** training regime (BabyLM-style; 10 epochs over the 10M-token corpus) and a **long** training regime. The long training regime of Charpentier & Samuel (2024) runs for 7812 steps with an average batch size of around 2M tokens (amounting to an estimated ∼1000 epochs). Due to computational constraints, our long training regime runs for 5000 steps with an average batch size of 1.1M tokens (amounting to about ∼350 epochs and requiring 96 GPU hours). We find that, even for such small models and training data, extended large-batch training boosts performance, which accounts for the difference in our reported GPT-BERT baseline performance on BLiMP (79.2%) and that of Charpentier & Samuel (2024) (81.2%).

For all short-regime training we report mean ± standard deviation over 3 random seeds. Since syntactic benchmark performance is known to be sensitive to logit temperature, we follow Charpentier & Samuel (2024) in reporting performance best temperature for each model.

## 4.2. Syntactic benchmarks

BLiMP evaluates syntax via minimal pairs, scoring whether a model prefers the grammatical sentence over a minimally different ungrammatical one. Recently, BLiMP performance of small models like GPT-BERT (81.2% with 10M tokens, 86.1% with 100M tokens) has come in range of much larger models like GPT-2 (83.0%) trained on 20B-30B tokens, while still falling short of human performance (88.6%) (Charpentier & Samuel, 2024; Warstadt et al., 2020).

SyntaxGym test suites are scored by surprisal over specified sentence regions and by whether the model satisfies a set of 2×2 inequality constraints over pairs of minimal pairs, making them less susceptible to global plausibility cues and generally more difficult (GPT-2: ∼78%). A random baseline scores ∼25% depending on the combination of suites used. We use 25 of the 31 suites, excluding the six Garden-Path Effects suites which test human-like ambiguity resolution rather than grammaticality (Hu et al., 2020).

Table 2 reports BLiMP and SyntaxGym performance for baseline vs. SAMBAL under short and long training, with breakdowns of performance by suite in Appendix B. SAMBAL matches the baseline in both regimes, while aggregate

*Table 2.* Syntactic benchmark performance (test accuracy %). SyntaxGym$^{\neg\text{refl}}$ removes reflexive suites.

| Model | BLiMP | SG | SG$^{\neg\text{refl}}$ |
|---|---|---|---|
| baseline (long) | 79.2 | 70.47 | 78.6 |
| SAMBAL (long) | 79.3 | 66.0 | 79.8 |
| baseline (short) | 71.2 ± 0.7 | 54.7 ± 2.0 | 63.7 ± 7.8 |
| SAMBAL (short) | 71.2 ± 0.8 | 46.6 ± 4.4 | 55.9 ± 4.9 |

SyntaxGym performances shows a gap (66.0 vs. 70.47 in the long training regime); . This discrepancy is concentrated in reflexive suites; removing these yields parity (79.8 for SAMBAL vs. 78.6 for baseline). We investigate the cause in Appendix D and summarize here:

**SAMBAL behavior on SyntaxGym reflexives.** SAMBAL reflexive suite scores collapse to near zero, and even the base model shows a large masculine–feminine gap (e.g., `reflexive_prep`: 57.9% masc vs. 5.3% fem; `reflexive_src`: 68.4% masc vs. 26.3% fem). A closer look shows that SAMBAL's collapse is driven by one recurring pattern in the singular condition: in examples like *The author that liked the senators hurt ___*, where the true antecedent (*author*) is singular, SAMBAL often prefers *themselves* over *himself/herself*. Because SyntaxGym counts a suite as correct only when all of its expected comparisons go in the right direction, this systematic error is sufficient to drive near-zero suite accuracy.

Targeted BLiMP-style diagnostics indicate that this is not a general inability to use reflexives: SAMBAL is near-ceiling when there is no distractor noun (e.g., *The manager congratulated herself/themselves*). Instead, the degradation is selective and emerges when a singular controller must be tracked across an intervening plural noun. The specific singular-controller + plural-distractor + reflexive configuration (and its reverse) that these suites test is essentially absent from our small corpus, and reflexives are skewed 2:1 towards the masculine *himself*. We interpret the base model's large masc–fem asymmetry as consistent with a near-threshold learning regime: there appears to be just enough evidence to partially learn the stronger generalization for masculine cases, but not enough to do so reliably for feminine cases. Small targeted augmentation largely repairs reflexive accuracy (Appendix D), indicating that the failure reflects missing coverage rather than a structural limitation.

## 4.3. Semantic suppression

We evaluate world-knowledge behavior with EWoK (Elements of World Knowledge), which probes basic world knowledge using minimal pairs from 11 domains. For example: "The piano is in front of Ali. Ali turns {left/right}...The piano is {right/left} of Ali." Even large LMs (1B–70B

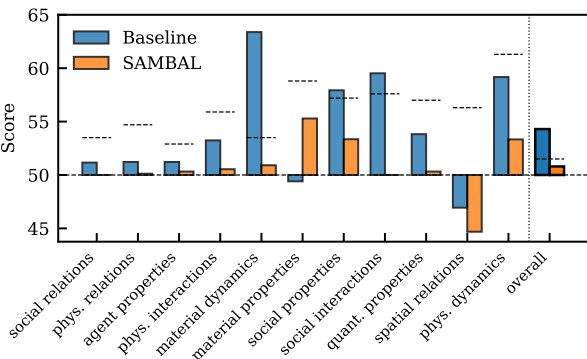

*Figure 2.* **World-knowledge probe (EWoK).** Accuracy by suite for SAMBAL vs. baseline. The dashed line indicates chance (50%); dotted reference lines show a 95% confidence threshold under a binomial test.

parameters) were found to substantially trail human performance (Ivanova et al., 2025). The baseline model attains only 54.3%, while SAMBAL scores at chance level (50.8%). Despite the small gap in overall performance, however, per-suite accuracies (Figure 2) reveal that the baseline scores above the 95% confidence level on 3 suites and in aggregate, while SAMBAL does on zero suites.

**Swap-probe minimal pairs: four-slot swaps.** To isolate plausibility/knowledge from grammatical well-formedness in an easier, tightly controlled setting, we construct a 100-sentence swap probe where each item is defined by four content slots $x_1, y_1, x_2, y_2$ instantiated into a fixed syntactic template. Swapping $y_1$ and $y_2$ (or $x_1$ and $x_2$) keeps words in the same syntactic roles and therefore preserves grammaticality, but is designed to disrupt plausibility, yielding a grammatical-implausible configuration (GI). Swapping $x_1 \leftrightarrow y_1$ and/or $x_2 \leftrightarrow y_2$ exchanges words across roles and yields ungrammatical-plausible (UP) and ungrammatical-implausible (UI) configurations. This construction helps to control for model sensitivity to word frequency and position (see Appendix F). For example:

| | |
|---|---|
| **template:** | *The boy $x_1$ the $y_1$ and $x_2$ the $y_2$.* |
| **vars:** | $x_1$: sails; $y_1$: boat; $x_2$: catches; $y_2$: mouse |
| **GP:** | *The boy sails the boat and catches the mouse.* |
| **GI:** | *The boy sails the mouse and catches the boat.* |
| **UP:** | *The boy boat the sails and mouse the catches.* |

Figure 1 plots the syntax logit margin $\Delta G = \log P(\text{grammatical}) - \log P(\text{ungrammatical})$ vs. the semantics logit margin $\Delta K = \log P(\text{plausible}) - \log P(\text{implausible})$. SAMBAL collapses $\Delta K$ toward zero, but strongly prefers the grammatical sentences, while the

*Table 3.* **Swap probe: syntax vs. semantic-plausibility preferences.** Accuracy and mean margins for syntax ($\Delta G$) and semantic plausibility ($\Delta K$).

| Model | syntax | | semantic plausibility | |
|---|---|---|---|---|
| | Acc. (%) | $\Delta G$ | Acc. (%) | $\Delta K$ |
| baseline (Long) | 55.0 | 0.65 | 100.0 | 7.62 |
| SAMBAL (Long) | 97.0 | 5.73 | 55.0 | 0.05 |

baseline shows large plausibility margins and no preference for grammaticality. Table 3 summarizes the same phenomenon as accuracy and mean margins, where accuracy is the fraction of templates with positive $\Delta$.

**Natural language inference (SNLI).** A standard benchmark of semantic capability, SNLI (Bowman et al., 2015) tests the ability to label a premise–hypothesis pair as entailment, contradiction, or neutral. However, the task conflates the compositional and structural semantics that SAMBAL preserves with the lexical and world knowledge it removes, and carries well-documented annotation artifacts and shallow shortcuts (Gururangan et al., 2018). Some pairs are resolvable largely from structure—e.g., dropping a conjunct, *"A woman is looking into a mirror, brushing her hair"* entails *"A woman is brushing her hair"*—whereas others require lexical or world knowledge—e.g., *"A soccer game with multiple males playing"* entails *"Some men are playing a sport"* only if one knows that soccer is a sport. Therefore, we do not treat it as a primary evaluation of semantics, and report full results in Appendix E.

### 4.4. Syntax–semantics disentanglement

**Minimal-pair conflict benchmark construction.** We test a regime where *grammaticality conflicts with an obviously plausible event interpretation.* We build a targeted grammar–plausibility *conflict set* (N=186) of minimal pairs in which the two sentences contain the same content words but differ only by swapping two noun phrases, so unigram frequency and lexical content are held constant. By construction, one member is grammatical but describes an implausible role assignment (*"The cake eats the children"*), while the other is plausible but contains a local agreement error (*"The children eats the cake"*).

To avoid confounding "entanglement" with intrinsically difficult constructions, we restrict to three local phenomena that both models handle well on BLiMP: (i) regular subject–verb agreement, (ii) determiner–noun number agreement, and (iii) passive auxiliary agreement (was/were). Each conflict item is accompanied by a matched *control* minimal pair that removes the conflict while keeping the same local grammatical dependency.

We retain items where (i) both models solve the control pair

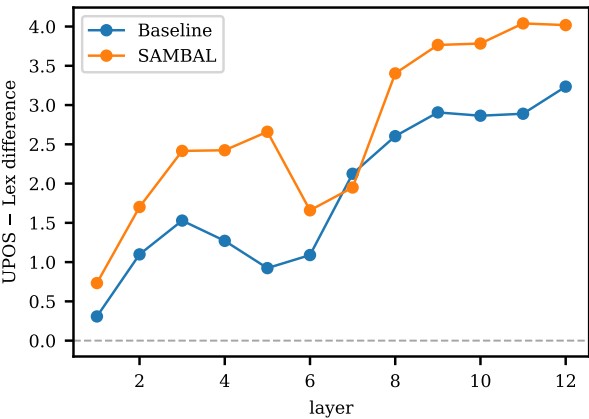

*Figure 3.* **Per-layer InfoNCE alignment (UPOS − Lex).** SAM-BAL shows a larger syntactic-over-lexical margin across layers, consistent with reduced lexical memorization pressure.

with a small positive margin (ii) and the baseline exhibits at least a mild plausibility preference on the conflict pair. Importantly, SAMBAL's behavior on the *conflict* comparison is *not* used for selection; it is only used to ensure that the control dependency is solvable. Because this procedure conditions on the presence of a plausibility lure for the baseline, the conflict set is baseline-conditioned by construction and should be read as an adversarial stress test of grammar–plausibility competition, not a general-purpose syntax benchmark. Our broader conclusions do not rest on it alone: the swap probe of Figure 1 and the SNLI results of §4.3 are not baseline-conditioned and point in the same direction. Further details on the benchmark are given in Appendix F.

**Results.** For evaluation, we combine the control and conflict sets and report the accuracy at best temperature. Both models maintain 100% accuracy on the control set, and the baseline (by design) attains only 10.2% accuracy on the conflict set, selecting the plausible member over the grammatical one on 89.8% of pairs. In contrast, SAMBAL achieves a striking 100% accuracy on the conflict set.

**Representation analysis with InfoNCE.** Behavioral probes suggest SAMBAL reduces plausibility reliance; we test whether this also appears in internal representations. Specifically, we ask whether hidden states become *relatively more aligned* with syntactic features (e.g., UPOS) than with lexical identity.

We compute token representations on the gold dependency parses from Universal Dependencies English Web Treebank corpus (Nivre et al., 2020). We quantify representational alignment using an InfoNCE-style contrastive score (van den Oord et al., 2018; Poole et al., 2019) in a supervised-contrastive setup where positives share a label

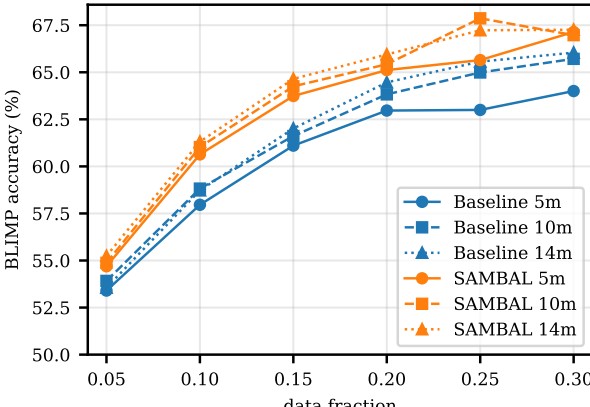

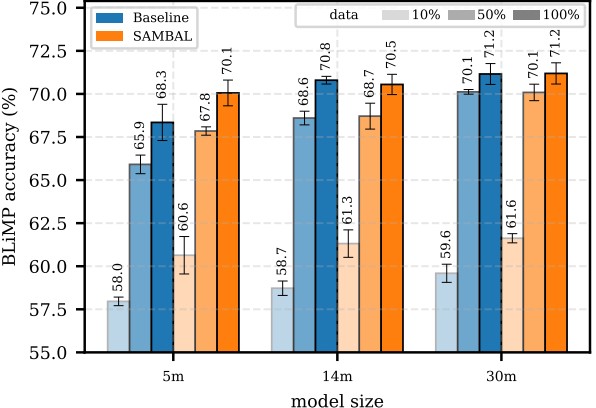

*Figure 4.* **Parameter and token efficiency on BLiMP. Top:** low-budget scaling curves up to 30% of the corpus for 5M/10M/14M models. **Bottom:** BLiMP vs. model size and data fraction (10%, 50%, 100%) for 5M/14M/30M models.

(Gunel et al., 2021). InfoNCE-like metrics have been used to characterize layerwise representation structure in LMs (Skean et al., 2024). We compute alignment scores for predicting (a) UPOS tags and (b) lexical identity.

Figure 3 plots the per-layer difference (UPOS alignment − Lex alignment) difference; larger values indicate stronger emphasis on syntax relative to lexical identity. We find that this syntax-over-lexical signal increases with depth in both models, consistent with higher layers encoding more abstract structure. However, SAMBAL shows a persistent upward shift: across layers, the UPOS−Lex gap is consistently larger by roughly 25% relative to the baseline, indicating that its hidden states are systematically more aligned with syntactic categories and less dominated by lexical identity.

### 4.5. Data & parameter scaling

Figures 4–5 summarize an efficiency sweep over data budget and model size. In the low-budget regime (up to 30% of the corpus; Fig. 4 top), SAMBAL consistently shifts BLiMP

scaling curves upward for 5M/10M/14M models, indicating improved per-token syntax learning. For example, for a 5M-parameter model, BLiMP increases 58.0→60.6 at 10% data and 64.0→67.1 at 30% data (corresponding means and standard deviations over 3 seeds are given in Appendix Fig. 5). Moreover, Fig. 4 (bottom) shows that these data-efficiency gains are concentrated in the smallest model and can persist even at higher budgets: for 5M , SAMBAL improves from 65.9→67.8 at 50% data and from 68.3→70.1 at 100% data), whereas once the model/data are large enough (14M/30M at 50–100%) the gap largely vanishes.

These gains also translate into *per-parameter* efficiency (Fig. 4 bottom): at the same 10% budget, a 5M SAM-BAL model slightly exceeds a 30M baseline (60.6 vs. 59.6). Again, the advantage largely disappears at higher budgets and larger models, consistent with SAMBAL helping most when data and/or capacity are limiting.

**Relative resource scarcity vs. absolute scale.** While our experiments are small scale, we have some indication that efficiency benefits may be more linked to relative scarcity rather than absolute scale. Relating SAMBAL's BLiMP advantage to a *data-utilization* measure (the baseline's accuracy as a fraction of the best SAMBAL accuracy at the same budget), the advantage is strongly predicted by how much headroom remains ($R^2 = 0.76$), and points from different data budgets intermingle rather than stratifying by absolute token count: the gain appears whenever the model is parameter-constrained *relative* to its data budget (Appendix G).

### 4.6. Small-domain adaptation

Can a SAMBAL model rapidly acquire domain content after pretraining? The use case is modular: SAMBAL pretraining encodes syntax in the base weights, and a lightweight LoRA adapter adds domain knowledge (only the adapter is trained)—targeting cases where one wants a linguistically-capable model that knows little beyond a target domain too small to train from scratch. We perform LoRA fine-tuning on the *Lord of the Rings* trilogy (580k words) (Hu et al., 2022). As expected, SAMBAL begins with much worse domain perplexity than baseline (190.4 vs. 64.6) but adapts quickly with limited exposure (36.3 vs. 29.2 after fine-tuning). The best model trained from-scratch on LoTR obtains PPL 55.5, indicating that specialization benefits from a syntax-competent initialization. Table 4 compares LoTR perplexity and BLiMP accuracy before vs. after specialization; we see that both SAMBAL and baseline models lose around 7% acc on BLiMP after fine-tuning. For comparison, the model trained on LoTR alone achieves only 53.9% on BLiMP.

Similar results hold for a more technical domain of the

*Table 4.* Specialization on LoTR with LoRA fine-tuning.

| Model | LoTR PPL | BLiMP Acc. |
|---|---|---|
| baseline (Long) | 64.6 | 79.2 |
| baseline (Long) fine-tuned | 29.2 | 72.4 |
| SAMBAL (Long) | 190.4 | 79.3 |
| SAMBAL (Long) fine-tuned | 36.3 | 71.4 |
| Best LoTR from-scratch model | 55.5 | 53.9 |

same size—a collection of PubMed abstracts (NLM, 2024) (Appendix J).

**LoTR generations.** Qualitatively, generations reflect the same pattern. Before specialization, SAMBAL tends to produce grammatical but semantically unusual continuations; after LoRA fine-tuning it begins to use domain-appropriate entities, themes and stylistic cues:

*'It's a dangerous business, Frodo, going out of your door' he used to say.*

**SAMBAL:** *Marlon went out comfortably before seeing, "How does that find our media?" To which the husband survived curiously and said, "Well, well...the one thing I meet is how we look and find with each other. When someone reaches personal surgery at a potato year...*

**SAMBAL ft:** *'We shall be too good here now.' The hobbits looked at him sadly and silently. Sam sat up, his face grappled with feeling that the hobbit was forgetting the sound of eating and singing softly.*

We include generations of the baseline model and additional SAMBAL generations in Appendix I.

## 5. Discussion

**Limitations, leakage, and ablations.** Even with constrained relexicalization, some semantic traces may remain in SAMBAL text (e.g., function-word distributions, constructions correlated with meaning, or distributional regularities induced by replacement statistics). We also preserve certain boundary features such as gender/humanness/animacy when they directly affect acceptability (agreement and binding), even though they correlate with meaning. The ablations clarify the role of these choices: allowing gender to be ablated reduces SAMBAL-short BLiMP from 71.2 to 70.47 and instead humanness ablation reduces it to 69.86; in reflexive/anaphor gender-agreement settings, dropping gender collapses behavior to approximately chance. At the same time, removing corpus-conditioned sampling drops BLiMP from 71.2 to 66.42, indicating that context-conditioned candidate selection is important for grammatical fit and coverage.

All of these components—residual function-word and constructional patterns, the retention of meaning-adjacent

boundary features (gender/humanness), and corpus-conditioned candidate selection—are plausible pathways by which distributional world-knowledge regularities could re-enter despite constrained relexicalization. SAMBAL is thus more precisely described as preserving syntax plus a thin layer of meaning-adjacent information than as a perfectly clean syntax/semantics separation; we view the delineation of this boundary—what to preserve, what to ablate, and why (Table 1, Appendix K)—as itself a contribution. Our no-knowledge probes indicate that plausibility and encyclopedic cues are substantially suppressed, but they do not rule out subtler leakage channels. A natural direction for future work is adversarial relexicalization: constructing transformations that intentionally reintroduce spurious form–semantics correlations between form and latent semantics/domain while preserving grammaticality, to better characterize when SAMBAL succeeds or fails at suppressing non-syntactic cues.

Another limiting factor is SAMBAL's reliance on high-resource English tooling (parsing, morphology, external lexicons), and these resources can be incomplete or idiosyncratic: permissive matching admits odd outputs, while strict matching can yield empty candidate pools. Some constructions we wish to control lack reliable off-the-shelf inventories (e.g., NPI licensors or fine-grained *to*-infinitival verb/adjective classes), motivating custom lexica and bucket definitions; portability to lower-resource languages remains open.

**Why might SAMBAL improve efficiency?** A working hypothesis is that SAMBAL improves data/compute efficiency by suppressing selectional preferences and plausibility priors, shifting training pressure toward morphosyntax, licensing, and structural dependencies. Future work could test this by mapping which phenomena benefit most and by varying the availability of semantic signals while holding grammatical constraints fixed.

Another possible source of learning advantage is the choice of replacement vocabulary: a large vocabulary could either help regularize learning or make it too challenging, and a small replacement vocabulary could simplify the task by shrinking the effective lexical space the model must model while acquiring syntactic constraints. We observe sensitivity to replacement vocabulary choice, exposing a tradeoff between syntactic isolation and lexical coverage. At one extreme, collapsing each class to a single representative could maximize structure-only training, but would harm real-vocabulary evaluations and downstream adaptation. Consistent with this, preliminary experiments restricting replacements to the small BLiMP vocabulary itself improve SAMBAL-short BLiMP by about 2 points, while decreasing SyntaxGym by about 2 points and increasing perplexity after LoTR adaptation. This suggests that down-stream tasks and adaptation metrics are useful controls to ensure that improvements on form-focused benchmarks are not achieved by drifting toward an overly simplified lexical distribution. Preliminary evidence shows that the efficiency advantage is not merely a lexical-entropy effect: a control trained on original text filtered to SAMBAL's top-25k vocabulary—preserving meaning but matching lexical entropy—lands above the baseline but below SAMBAL at the 10% token budget, so lower entropy explains part but not all of it (Appendix H). Systematically exploring the choice of replacement vocabulary—and tracing out grammar/coverage/adaptation tradeoffs as replacement vocabularies are tightened or broadened—is a promising direction for future work.

## Impact Statement

This paper presents work whose goal is to advance the field of Machine Learning. There are many potential societal consequences of our work, none which we feel must be specifically highlighted here.

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

# A. Implementation details for SAMBAL

SAMBAL transforms natural sentences into *grammatical nonsense* via constrained relexicalization: we replace eligible *open-class* tokens (content nouns/verbs/adjectives/adverbs and proper nouns) with new lemmas drawn from syntactically compatible candidate pools, then inflect the sampled lemma to match the original token's morphosyntactic realization. The design goal is to preserve grammatical structure (agreement, morphosyntax, and coarse complement structure / c-selection) while ablating semantic and world-knowledge content.

**Primary configuration used in our experiments.**  Unless explicitly stated otherwise, we run SAMBAL with the following default settings:

- **Parser / tagger:** a transformer-based English pipeline (spaCy en_core_web_trf (Honnibal et al., 2020)) provides tokenization, POS tags (PTB), morphology, and dependencies.

- **Replacement scope:** proper nouns are replaced; pronouns are preserved. Closed-class tokens are preserved (see below).

- **Humanness and gender preservation:** Human-denoting heads are replaced with other human-denoting heads, and gender-marked words/names are replaced within the same gender class.

- **Vocabulary gating:** swap candidates with a minimum frequency threshold of 1 in the training corpus are allowed.

- **Context-conditioned sampling:** uses frequency-weighted sampling from corpus-derived context buckets, with (i) a per-bucket minimum lemma count of 2 and a backoff policy requiring at least 2 surviving candidates and at least 5% of the unconditioned pool before accepting a bucket.

- **To-infinitive safety:** In raising/control/ECM-sensitive infinitival environments, we restrict replacements using dedicated lexicons; if no safe compatible pool is available, we freeze the governing head rather than falling back to generic sampling.

- **Round-trip validation:** We reparse the augmented output and reject–resample 4 times if structural invariants fail (details below).

- **VerbNet soft-preposition fallback:** enabled with a minimum pool size threshold of 10. For plain intransitive verbs with adjunct PPs, if enforcing the observed governed preposition would shrink the VerbNet candidate set below 10, we ignore the preposition in VerbNet matching (while still retaining it as a contextual feature for corpus-based gating).

**External resources and derived inventories.**  SAMBAL relies on a set of static inventories and lexicon:

- **WordNet:** provides large lemma inventories for nouns, adjectives, and adverbs (used as base replacement pools) and supports additional coarse syntactic checks for verbs (e.g., minimum object requirements and complement-type compatibility) (Miller, 1995)

- **VerbNet:** provides verb classes and frame information used to bucket verbs by coarse subcategorization signatures.

- **Countability lexicon for nouns derived from Wiktionary:** supplies coarse noun classes *mass*, *count*, *both*, and *plural-only* to preserve determiner/number compatibility (see below). (Wiktionary)

- **Function-word list (UD English Web Treebank):** extracted from UD-EWT tokens whose universal POS is in {ADP, DET, AUX, PRON, PART, SCONJ, CCONJ}, providing a high-coverage function-word inventory beyond stopword heuristics. (Universal Dependencies, 2023)

- **Multiword-expression (MWE) inventories:** (i) fixed MWEs harvested from UD-EWT fixed relations, (ii) verb–particle constructions from UD-EWT (compound:prt), and (iii) verbal MWE patterns derived from STREUSLE (compiled into matcher patterns). These spans are preserved as atomic units during augmentation. (Universal Dependencies, 2023; Schneider & Smith, 2018)

- **Negative polarity items (NPIs):** a Wiktionary-derived list of English NPIs is used both to (i) preserve NPI tokens themselves and (ii) prevent NPIs from being introduced as replacement candidates.

- **NPI licensor patterns:** a hand-curated inventory of token- and dependency-based licensor patterns marks additional protected spans (e.g., negation-like and quantificational licensors).

- **Human noun inventory:** derived from WordNet `noun.person` (plus associated multiword human expressions), enabling human→human replacement.

- **Names and proper nouns:** U.S. SSA baby names are used to construct given-name pools partitioned into male/female/neutral (based on aggregate frequency and dominance criteria). We additionally maintain a training-corpus-frequency derived inventory of allowable single-token proper nouns for replacement.

- **Gendered common words:** a filtered version of the `ecmonsen/gendered_words` lexicon supplies unambiguous gender labels for common nouns; we prefilter it against the allowed vocabulary. (ecmonsen, 2023)

- **Allowed vocabulary:** an inventory of surface tokens extracted from the target pretraining corpus using the same tokenizer/parser, used to gate substitutions so that realized outputs stay in-vocabulary.

- **Context statistics:** corpus-derived counts of lemma occurrences conditioned on syntactic context buckets, used for context-conditioned gating and frequency-weighted sampling. We build these statistics by parsing the corpus and tabulating lemma counts keyed by compact context signatures; to control size we restrict collection to a fixed high-frequency word list, the top-25k words according to wordfreq python library(wordfreq developers, 2022).

**Protected tokens and spans.** Before sampling any replacements, we mark protected items and leave them unchanged. Protection includes: (i) closed-class tokens (function words, auxiliaries, complementizers, determiners, adpositions, conjunctions, etc.), (ii) any token in the function-word inventory or stopword set, (iii) NPI tokens and detected licensor spans (both token-based and dependency-pattern matches), (iv) any token inside a detected MWE span (fixed MWEs and verbal MWEs), and (v) structural backbones such as lexical *be* and auxiliary/copular heads in sensitive configurations. We also conservatively protect tokens that behave as grammar-critical operators in minimal-pair settings (e.g., quantificational/degree constructions), including a WordNet-derived set of quantity nouns used to freeze binominal degree MWEs (e.g., *a lot of*, *a number of*).

**Replacement pools and constraints by category.** Replacements are defined at the *lemma* level; we then realize a surface form that matches the original token's PTB tag and morphosyntactic profile.

**Nouns (common nouns).** We bucket noun lemmas by coarse countability class. At runtime, we select a noun pool compatible with the slot's number and determiner context:

- *Bare singular* heads (singular common nouns without an overt determiner) are only replaced with lemmas that can appear bare (typically mass-capable or otherwise licensed).

- *Plural* heads are replaced with lemmas compatible with plural realization, including plural-only nouns.

- *Determiner-sensitive* contexts respect mass/count behavior so that replacements do not force impossible determiner patterns.

When humanness preservation is enabled, human-denoting noun heads are replaced from a human-only pool; multiword human expressions are preserved as spans.

**Proper nouns.** We replace each eligible proper-noun token with a sampled name while preserving casing. If the token matches a known given name with a gender label, we sample a replacement from the same gender pool; otherwise we sample from the allowed-proper-noun inventory.

**Verbs.** We compute a coarse *subcategorization signature* from the dependency parse and retrieve verb candidates from VerbNet buckets keyed by:

- *Objecthood:* intransitive vs. transitive vs. ditransitive structure (with passives handled explicitly).

- *Complement kind:* none vs. NP object vs. double-object vs. clausal complements (that/wh), infinitival complements (to/for-to), and gerundive complements.

- *Governed prepositions:* observed governed prepositions are treated as selectional constraints when they plausibly reflect argument structure; for adjunct-like PP attachments under plain intransitives we apply the soft-preposition fallback described above.

We then apply conservative additional guards to avoid known failure modes: we freeze auxiliaries/copulas and *be*, avoid swapping heads in expletive/ECM-like configurations, and freeze phrasal-verb heads (particle compatibility is not guaranteed). We further filter candidates using WordNet-derived syntactic-frame checks for (i) minimum object requirements in transitive slots, (ii) compatibility with clause and infinitival complement types, and (iii) compatibility with specific PP head types when required.

**To-infinitive control/raising/ECM environments (optional but enabled by default).** Some heads differ systematically in whether and how they license infinitival complements and argument realization. When a verb or adjective participates in a raising/control/ECM-sensitive infinitival spine, we restrict candidates using dedicated lexicons that encode a small set of *license tags* (e.g., subject-control vs. object-control vs. raising vs. ECM, plus expletive *it/there* and passive variants). Only lemmas whose licenses subsume the local surface pattern are eligible. If a compatible pool cannot be formed, we freeze the governing head (rather than falling back to generic VerbNet/WordNet sampling).

**Adjectives.** Adjective candidates are drawn from a WordNet-derived adjective inventory, filtered to avoid proper/demonym-like items. In infinitival-adjective spines (e.g., *easy to V*), we again apply license-based filtering using the to-infinitive adjective lexicon, including special handling for stranded-preposition ("gap-PP") patterns.

**Adverbs.** Adverb candidates come from a WordNet-derived adverb inventory that is prefiltered to items that tag as plain adverbs (RB) under the parser's tagger (to avoid degree-morphology edge cases by default).

**Context-conditioned candidate gating and sampling.** To reduce parser-fragility and improve surface plausibility without reintroducing semantic content, we condition replacements on corpus-derived syntactic context statistics. For each replaceable token, we compute a compact *context bucket key* composed of: (i) POS and PTB tag, (ii) dependency role and head POS/tag, (iii) local neighborhood cues (nearby POS tags and a coarse relative-position feature), (iv) a compact morphological signature, and (v) verb-specific frame features (coarse valency, complement kind, and governed-preposition set). We intersect the lexicon-derived candidate set with the set of lemmas observed in that context bucket in the base corpus and sample replacements proportionally to their observed bucket frequencies. If the intersection is too small (absolute and relative thresholds given in the primary configuration above), we back off through a deterministic chain of progressively coarser buckets (dropping some contextual features) until we obtain a sufficiently large candidate set.

**Surface realization and post-processing.** After sampling a replacement lemma, we inflect it to match the original token's PTB tag using a morphology engine (lemmatization/inflection for nouns and verbs, with conservative fallbacks). We preserve orthographic casing by matching the original token's capitalization pattern. Finally, we apply a local determiners fix for *a/an* to ensure that the indefinite article matches the pronunciation of the following word after substitution.

**Round-trip validation (UD reparse).** When enabled, we reparse the augmented output and require it to preserve a small set of robust, grammar-relevant invariants. Concretely, we compare: (i) the multiset of coarse verb-frame keys (intrans/trans/ditrans), (ii) the counts of expletive constructions (*there* vs. *it*), (iii) the counts of tough-like vs. raising-like infinitival adjective patterns under copular *be*, and (iv) a conservative count of ECM/expletive-raising configurations. If any invariant differs, we reject the sample and resample replacements (up to the configured maximum number of retries), accepting the first augmentation that passes the checks.

**Parser quality and manual validation.** Because relexicalization is conditioned on the initial parse, tagging and dependency errors can propagate downstream. The `en_core_web_trf` pipeline reports high accuracy (roughly 0.98 POS, 0.95 UAS, 0.94 LAS) (Honnibal et al., 2020), and our pipeline introduces occasional realization errors of its own. To gauge the net fidelity of the transformed corpus, we manually spot-checked 100 sampled SAMBAL sentences and found 87% to be fully grammatical. Despite this imperfection, we take the downstream BLiMP and SyntaxGym results as evidence that this level of fidelity is sufficient for our purposes.

# B. Full BLiMP and SyntaxGym suite results

*Table 5.* BLiMP per-suite accuracy (%).

| BLiMP paradigm | baseline | SAMBAL |
|---|---|---|
| ANAPHOR AGREEMENT | | |
| anaphor_gender_agreement | 89.7 | 94.1 |
| anaphor_number_agreement | 94.9 | 96.6 |
| ARGUMENT STRUCTURE | | |
| animate_subject_passive | 78.6 | 77.2 |
| animate_subject_trans | 87.3 | 88.8 |
| causative | 70.1 | 50.7 |
| drop_argument | 67.5 | 71.4 |
| inchoative | 68.4 | 58.1 |
| intransitive | 75.4 | 72.6 |
| passive_1 | 88.5 | 81.4 |
| passive_2 | 87.6 | 86.7 |
| transitive | 82.4 | 64.8 |
| BINDING | | |
| principle_A_c_command | 46.7 | 57.6 |
| principle_A_case_1 | 99.9 | 99.7 |
| principle_A_case_2 | 96.2 | 88.0 |
| principle_A_domain_1 | 84.4 | 91.1 |
| principle_A_domain_2 | 59.8 | 58.9 |
| principle_A_domain_3 | 55.9 | 59.4 |
| principle_A_reconstruction | 26.4 | 37.9 |
| CONTROL/RAISING | | |
| existential_there_object_raising | 81.3 | 63.7 |
| existential_there_subject_raising | 86.5 | 78.0 |
| expletive_it_object_raising | 66.9 | 59.6 |
| tough_vs_raising_1 | 74.4 | 66.2 |
| tough_vs_raising_2 | 78.0 | 81.4 |
| DETERMINER-NOUN AGR. | | |
| determiner_noun_agreement_1 | 98.7 | 97.0 |
| determiner_noun_agreement_2 | 99.1 | 98.0 |
| determiner_noun_agreement_irregular_1 | 89.1 | 89.7 |
| determiner_noun_agreement_irregular_2 | 96.4 | 93.7 |
| determiner_noun_agreement_with_adjective_1 | 96.5 | 94.6 |
| determiner_noun_agreement_with_adj_2 | 97.2 | 95.1 |
| determiner_noun_agreement_with_adj_irregular_1 | 84.5 | 90.7 |
| determiner_noun_agreement_with_adj_irregular_2 | 95.3 | 91.7 |
| ELLIPSIS | | |
| ellipsis_n_bar_1 | 76.6 | 76.9 |
| ellipsis_n_bar_2 | 96.3 | 74.1 |
| FILLER-GAP | | |
| wh_questions_object_gap | 70.9 | 74.4 |
| wh_questions_subject_gap | 83.5 | 81.5 |
| wh_questions_subject_gap_long_distance | 78.7 | 73.0 |
| wh_vs_that_no_gap | 90.2 | 87.2 |

*Continued on next page*

| BLiMP paradigm | baseline | SAMBAL |
|---|---|---|
| wh_vs_that_no_gap_long_distance | 94.1 | 93.8 |
| wh_vs_that_with_gap | 89.2 | 93.2 |
| wh_vs_that_with_gap_long_distance | 59.9 | 61.4 |
| IRREGULAR FORMS | | |
| irregular_past_participle_adjectives | 96.6 | 99.4 |
| irregular_past_participle_verbs | 86.5 | 85.7 |
| ISLAND EFFECTS | | |
| adjunct_island | 80.8 | 91.3 |
| complex_NP_island | 41.0 | 54.4 |
| coordinate_structure_constraint_complex_left_branch | 76.2 | 94.6 |
| coordinate_structure_constraint_object_extraction | 92.9 | 93.1 |
| left_branch_island_echo_question | 21.9 | 20.2 |
| left_branch_island_simple_question | 90.5 | 97.5 |
| sentential_subject_island | 35.0 | 50.1 |
| wh_island | 75.0 | 90.1 |
| NPI LICENSING | | |
| matrix_question_npi_licensor_present | 80.0 | 84.1 |
| npi_present_1 | 51.1 | 86.4 |
| npi_present_2 | 57.8 | 65.3 |
| only_npi_licensor_present | 95.3 | 99.0 |
| only_npi_scope | 85.5 | 82.6 |
| sentential_negation_npi_licensor_present | 98.3 | 94.8 |
| sentential_negation_npi_scope | 78.5 | 59.4 |
| QUANTIFIERS | | |
| existential_there_quantifiers_1 | 98.0 | 98.1 |
| existential_there_quantifiers_2 | 26.2 | 14.3 |
| superlative_quantifiers_1 | 57.7 | 100.0 |
| superlative_quantifiers_2 | 79.4 | 78.4 |
| SUBJECT-VERB AGR. | | |
| distractor_agreement_relational_noun | 91.2 | 88.5 |
| distractor_agreement_relative_clause | 82.6 | 85.3 |
| irregular_plural_subject_verb_agreement_1 | 94.4 | 87.2 |
| irregular_plural_subject_verb_agreement_2 | 90.9 | 87.1 |
| regular_plural_subject_verb_agreement_1 | 90.0 | 84.9 |
| regular_plural_subject_verb_agreement_2 | 90.1 | 93.8 |

*Table 6.* SyntaxGym per-suite accuracy (%)

| SyntaxGym suite | baseline | SAMBAL |
|---|---|---|
| number_orc | 47.37 | 68.42 |
| number_prep | 63.16 | 84.21 |
| number_src | 68.42 | 57.89 |
| reflexive_orc_fem | 15.79 | 0.00 |
| reflexive_orc_masc | 31.58 | 0.00 |
| reflexive_prep_fem | 5.26 | 0.00 |
| reflexive_prep_masc | 57.89 | 5.26 |
| reflexive_src_fem | 26.32 | 0.00 |
| reflexive_src_masc | 68.42 | 0.00 |
| npi_orc_any | 86.84 | 89.47 |
| npi_orc_ever | 94.74 | 94.74 |
| npi_src_any | 65.79 | 84.21 |
| npi_src_ever | 100.00 | 97.37 |
| fgd_hierarchy | 20.83 | 10.42 |
| fgd_object | 66.67 | 79.17 |
| fgd_pp | 37.50 | 54.17 |
| fgd_subject | 37.50 | 45.83 |
| center_embed | 85.71 | 64.29 |
| center_embed_mod | 75.00 | 57.14 |
| cleft | 97.50 | 100.00 |
| cleft_modifier | 97.50 | 95.00 |
| subordination | 100.00 | 95.65 |
| subordination_orc-orc | 100.00 | 100.00 |
| subordination_pp-pp | 100.00 | 95.65 |
| subordination_src-src | 95.65 | 95.65 |

## C. Ablation table

*Table 7.* BLiMP ablation results.

| Ablation | BLiMP |
|---|---|
| SAMBAL (Short) | $71.2 \pm 0.8$ |
| no to-inf | $70.65 \pm 0.30$ |
| no gender | $70.47 \pm 0.24$ |
| no UD-roundtrip | $70.38 \pm 0.42$ |
| no humanness | $69.86 \pm 0.86$ |
| no context buckets | $66.42 \pm 0.78$ |

## D. SAMBAL reflexive performance exploration and diagnosis

**What goes wrong.** SyntaxGym reflexive suites are an outlier in our evaluation: SAMBAL collapses to near-zero suite accuracy, and even the base model shows a large masculine–feminine gap (e.g., `reflexive_prep`: 57.9% masc vs. 5.3% fem; `reflexive_src`: 68.4% masc vs. 26.3% fem). Inspecting condition-level behavior, we found that SAMBAL's collapse is driven by a single recurring error in the *singular* condition: in items like *The author that liked the senators hurt __*, where the true antecedent (*author*) is singular but a plural noun intervenes (*senators*), SAMBAL often prefers *themselves* over *himself/herself*. Because SyntaxGym suites require all expected comparisons to hold simultaneously, consistently failing this singular contrast is sufficient to drive near-zero suite scores.

**Targeted BLiMP-style diagnostics.** To localize the failure, we constructed BLiMP-format minimal pairs that separate interference from baseline lexical preferences. The results show a sharply selective deficit: (i) SAMBAL is near-ceiling when interference is removed (100% on singular-attractor controls), (ii) SAMBAL passes fixed-form licensing controls (e.g., *himself*-only and *themselves*-only number tests are both >90%), but (iii) SAMBAL drops substantially when a singular controller must be tracked across an intervening plural noun (32.22% on the plural-attractor diagnostic at T=1.0; 63.33% at the best temperature), matching the singular-contrast failure observed in SyntaxGym.

**Corpus-side evidence and why we treat this as near-threshold.** We examined if this failure can be traced to surface-level statistics of the training corpus. The precise singular-controller + plural-distractor + reflexive configuration (and its reverse) that the SyntaxGym suites test is essentially absent from our corpus: of 5,069 extracted reflexive occurrences, only 39 fall in the broad neighborhood of this pattern (singular controller, intervening plural noun, reflexive), and these belong to a wider family of cue-conflict cases that do not teach the exact tested contrast. This sparsity, together with the base model's already-large masc–fem asymmetry on SyntaxGym reflexives, is consistent with a near-threshold learning regime in which there is limited and uneven signal for the stronger long-distance generalization (enough to partially learn the masculine case, but not reliably for the feminine case). Corpus statistics for these patterns are very similar between the original and SAMBAL-ablated corpora, so we do not attribute SAMBAL reflexive performance to a large distributional shift; however, in such a data-sparse regime, even small degradations in effective learnability (e.g., slightly harder retrieval conditions or noisier controller tracking under ablation) could plausibly push performance over a learnability edge.

**Targeted augmentation repairs performance.** To test whether the collapse reflects a structural inability to learn long-distance binding or merely missing pattern coverage, we augmented the training corpus with relevant examples. Specifically, we created two SAMBAL-ablated sentences from each of the 228 grammatical SyntaxGym reflexive items and (because reflexive-only augmentation slightly hurt number-suite behavior) added analogous examples from the number-suite items as well, increasing the corpus by only about 6,000 tokens. Training short-regime models on the augmented SAMBAL corpus raises the reflexive-suite average to 98.0%, with overall SyntaxGym improving to $66.8 \pm 6.9$. Augmenting the baseline corpus with the same examples yields 100% on the reflexive suites and $73.8 \pm 1.3$ on SyntaxGym overall. We therefore interpret the collapse as a genuine weak point of the 10M-token regime that is driven by missing pattern coverage rather than by an inability of SAMBAL-trained models to learn long-distance binding in principle: the relevant pattern is adversarial and uncommon in natural corpora, the 10M-token corpus is near-threshold for it, and modest targeted augmentation largely repairs it.

# E. SNLI evaluation details

We evaluate on SNLI (Bowman et al., 2015), classifying whether a hypothesis is entailed by, contradicts, or is neutral to a premise (chance = 33.3%). Table 8 reports full-test and SNLI-hard accuracy. The above-chance SAMBAL score should be read against SNLI's well-documented shallow shortcuts: a hypothesis-only classifier reaches ~67% (Gururangan et al., 2018) and a word-overlap baseline 50.4% (Bowman et al., 2015). The artifact-controlled SNLI-hard subset (Gururangan et al., 2018) removes hypothesis-only artifacts; there the baseline obtains 44.3% and SAMBAL 37.5%—much closer to chance—while preserving the ~7pp SAMBAL–baseline gap seen on the full set, consistent with SAMBAL retaining little broad semantic competence beyond what shallow cues afford. The variant trained without context-conditioned sampling (59.7%) essentially matches standard SAMBAL (60.1%), indicating that context-conditioned sampling injects little semantic content.

*Table 8.* SNLI results (accuracy %). "no ctx. samp." is the SAMBAL variant without context-conditioned sampling.

| Model | SNLI | SNLI-hard |
|---|---|---|
| baseline | 67.4 | 44.3 |
| SAMBAL | 60.1 | 37.5 |
| SAMBAL (no ctx. samp.) | 59.7 | – |
| chance | 33.3 | 33.3 |

# F. Probe suite construction details

Here we detail how we construct and score the two probe sets used in §4.

- the *four-slot swap probe* (syntax vs. semantic plausibility preferences under a fixed template), and

- the *grammar–plausibility conflict benchmark* (role-reversal swap probes with local agreement errors).

### F.1. Swap-probe benchmark: four-slot swaps

**Goal.** The swap probe isolates a model's preference for (a) grammatical well-formedness vs. (b) semantic plausibility, while controlling lexical identity and (to a large extent) unigram frequency effects by reusing the same content words across derived variants.

**Template and slots.** Each item is defined by four open-class slots $(x_1, y_1, x_2, y_2)$ instantiated into a fixed syntactic template with two transitive predicates. We evaluate *both* conjunct orders to control for position/order effects:

$$T^{12}(a, b, c, d) = \texttt{The boy } a \texttt{ the } b \texttt{ and } c \texttt{ the } d.$$
$$T^{21}(a, b, c, d) = \texttt{The boy } c \texttt{ the } d \texttt{ and } a \texttt{ the } b.$$

Here $x_1, x_2$ are transitive verbs (surface-inflected to match the template), and $y_1, y_2$ are count nouns compatible with *the*. We construct items so that the base pairing $(x_1, y_1)$ and $(x_2, y_2)$ yields an intuitively plausible base sentence.

**Derived variants via swaps (with both conjunct orders).** From each slot quadruple we deterministically generate four configurations (GP, GI, UP, UI), and for each configuration we instantiate two sentence variants corresponding to the two conjunct orders (12 and 21). This yields 8 sentences per item.

1. **GP (grammatical–plausible).** The base instantiation:

$$\text{GP}^{12} = T^{12}(x_1, y_1, x_2, y_2), \qquad \text{GP}^{21} = T^{21}(x_1, y_1, x_2, y_2).$$

2. **GI (grammatical–implausible).** Swap the two objects, keeping words in their original syntactic roles:

$$\text{GI}^{12} = T^{12}(x_1, y_2, x_2, y_1), \qquad \text{GI}^{21} = T^{21}(x_1, y_2, x_2, y_1).$$

This preserves grammaticality by construction but is intended to disrupt plausibility by mismatching verb–object pairings.

3. **UP (ungrammatical–plausible).** Swap each verb with its corresponding object (crossing syntactic roles):

$$\mathrm{UP}^{12} = T^{12}(y_1, x_1, y_2, x_2), \qquad \mathrm{UP}^{21} = T^{21}(y_1, x_1, y_2, x_2).$$

This keeps the same multiset of content words and largely preserves the intended event associations at the bag-of-words level, while introducing a clear category/word-order violation.

4. **UI (ungrammatical–implausible).** Apply both perturbations (role-crossing swap plus object swap):

$$\mathrm{UI}^{12} = T^{12}(y_2, x_1, y_1, x_2), \qquad \mathrm{UI}^{21} = T^{21}(y_2, x_1, y_1, x_2).$$

**Order-averaged scores.** Let $s_m(\cdot)$ denote the sentence score for model $m$ under our pseudo-log-likelihood protocol (§4.1). For each configuration $C \in \{\mathrm{GP}, \mathrm{GI}, \mathrm{UP}, \mathrm{UI}\}$, we define an order-averaged score:

$$\bar{s}_m(C) \;=\; \tfrac{1}{2}\Big( s_m(C^{12}) + s_m(C^{21}) \Big).$$

This averaging reduces sensitivity to conjunct order and token-position effects.

**Margins and accuracies (averaging over the two order variants).** We define a *semantic plausibility* margin by comparing grammatical-plausible vs. grammatical-implausible, averaging over both conjunct orders:

$$\Delta_K^m \;=\; \bar{s}_m(\mathrm{GP}) - \bar{s}_m(\mathrm{GI}) \;=\; \tfrac{1}{2} \sum_{o \in \{12,21\}} s_m(\mathrm{GP}^o) \;-\; \tfrac{1}{2} \sum_{o \in \{12,21\}} s_m(\mathrm{GI}^o). \tag{1}$$

We define a *syntax/grammar* margin by comparing *all* grammatical sentences (GP and GI in both orders) against *all* ungrammatical sentences (UP and UI in both orders):

$$\Delta_G^m = \tfrac{1}{4} \sum_{o \in \{12,21\}} \Big( s_m(\mathrm{GP}^o) + s_m(\mathrm{GI}^o) \Big) \;-\; \tfrac{1}{4} \sum_{o \in \{12,21\}} \Big( s_m(\mathrm{UP}^o) + s_m(\mathrm{UI}^o) \Big) \tag{2}$$

$$= \tfrac{1}{2}\Big( \bar{s}_m(\mathrm{GP}) + \bar{s}_m(\mathrm{GI}) \Big) \;-\; \tfrac{1}{2}\Big( \bar{s}_m(\mathrm{UP}) + \bar{s}_m(\mathrm{UI}) \Big). \tag{3}$$

We report "plausibility accuracy" as $\mathbf{1}[\Delta_K^m > 0]$ averaged over items and "syntax accuracy" as $\mathbf{1}[\Delta_G^m > 0]$ averaged over items; we also report mean margins $\mathbb{E}[\Delta_K^m]$ and $\mathbb{E}[\Delta_G^m]$.

**Item selection.** We construct a fixed set of 100 slot-quadruples by choosing common transitive verbs and concrete nouns so that (i) the base sentence is intuitively plausible and (ii) the swapped-object variant is anomalous while remaining grammatical.

## F.2. Grammar–plausibility conflict benchmark: role-reversal swaps

**Goal.** This benchmark targets settings where grammatical well-formedness conflicts with obvious semantically plausible interpretation. Each item is a *swap-only* minimal pair: the two sentences contain the same content words and differ only by swapping two noun phrases, which controls unigram identity while flipping thematic roles. Grammaticality is determined locally by an agreement dependency.

**Template families (local, high-BLiMP phenomena).** To avoid confounding "entanglement" with intrinsically difficult constructions, we restrict to short templates built from three local phenomena that both models handle well on BLiMP:

1. **SVA role reversal (regular subject–verb agreement).** Let $A_{\mathrm{pl}}$ be an agent-like plural noun (e.g., *children*) and $B_{\mathrm{sg}}$ a patient-like singular noun (e.g., *cake*), and let $V_{3\mathrm{sg}}/V_{\mathrm{pl}}$ be the corresponding verb forms.

$$\text{GOOD (grammatical, implausible)} : \textit{The } B_{\mathrm{sg}} \; V_{3\mathrm{sg}} \textit{ the } A_{\mathrm{pl}}.$$
$$\text{BAD (ungrammatical, plausible)} : \textit{The } A_{\mathrm{pl}} \; V_{3\mathrm{sg}} \textit{ the } B_{\mathrm{sg}}.$$

2. **Det–noun role reversal (determiner–noun number agreement).** We use a singular-only determiner (e.g., *each* or *a*) and a flexible determiner (e.g., *some*) with a past-tense verb to keep verb morphology neutral:

$$\text{GOOD} : \textit{Each } B_{\text{sg}} \, V_{\text{past}} \textit{ some } A_{\text{pl}}.$$
$$\text{BAD} : \textit{Each } A_{\text{pl}} \, V_{\text{past}} \textit{ some } B_{\text{sg}}.$$

3. **Passive role reversal (auxiliary agreement in passives).** With past participle $V_{\text{pp}}$:

$$\text{GOOD} : \textit{The } A_{\text{pl}} \textit{ were } V_{\text{pp}} \textit{ by the } B_{\text{sg}}.$$
$$\text{BAD} : \textit{The } B_{\text{sg}} \textit{ were } V_{\text{pp}} \textit{ by the } A_{\text{pl}}.$$

In all families, the GOOD and BAD sentences differ only by swapping $A_{\text{pl}}$ and $B_{\text{sg}}$; grammaticality is determined by the agreement morphology ($V_{3sg}$ vs. plural subject, singular determiner vs. plural head, or *was/were*).

**Controls (to ensure the phenomenon is easy).** Each conflict item is paired with a matched *control* minimal pair that removes the grammar–plausibility conflict while keeping the same local agreement dependency. Concretely, we reuse the conflict BAD sentence as `control_bad`, and form `control_good` by keeping the *plausible* role assignment but correcting only the agreement morphology (e.g., *The cake was eaten by the children* vs. *The cake were eaten by the children*). This ensures that any errors on the conflict set are not simply due to failure to represent the underlying agreement rule.

**Candidate generation.** We generate a large pool of candidates by sampling from a small inventory of typed micro-frames: agent-like plural subjects (animate groups), patient-like singular objects (typically inanimate), and strongly agentive verbs with the required inflections ($V_{\text{pl}}, V_{3\text{sg}}, V_{\text{past}}, V_{\text{pp}}$). The grammatical label is determined purely by the template.

**Automatic scoring and filtering (stress-test slice).** We score all sentences with a pseudo-log-likelihood evaluation protocol with temperature=1. Let $\Delta^m_{\text{ctrl}} = s_m(\texttt{control\_good}) - s_m(\texttt{control\_bad})$ be the control margin for model $m$, and let $\Delta^{\text{base}}_{\text{conf}} = s_{\text{base}}(\text{BAD}) - s_{\text{base}}(\text{GOOD})$ be the baseline's preference for the plausible-but-ungrammatical member. We retain candidates that satisfy:

1. **Control solvability (both models):** $\Delta^{\text{base}}_{\text{ctrl}} \geq \tau$ and $\Delta^{\text{SAMBAL}}_{\text{ctrl}} \geq \tau$, where $\tau = 0.1$.

2. **Presence of a plausibility lure (baseline only):** $\Delta^{\text{base}}_{\text{conf}} \geq \gamma$, where $\gamma = 0.1$.

We then subsample a fixed number per family with simple de-duplication constraints (e.g., not repeating the same noun pair/verb frame excessively). Importantly, SAMBAL's behavior on the *conflict* comparison is not used for selection; it is used only to ensure that the control dependency is solvable. Because the filtering conditions on the existence of a plausibility lure for the baseline, the resulting conflict set should be interpreted as *susceptibility under adversarial tension* rather than as a general-purpose syntax benchmark.

## G. Additional scaling results

**Relative resource scarcity vs. absolute scale.** Figure 6 plots SAMBAL's BLiMP advantage (pp) against the data-utilization measure of §4.5: the baseline's accuracy as a percentage of the maximum SAMBAL accuracy at the same data budget. The 25 configurations span 8 data budgets (5%–100% of the corpus) and 4 model sizes (5M–30M parameters). The relationship is strongly linear ($R^2 = 0.76$, Pearson $r = -0.87$), with a slope of about $-0.47$pp per percentage point of utilization: configurations where the baseline sits below $\sim$97% of the SAMBAL ceiling consistently show $+1.5$ to $+3.2$pp advantage, while those at or above the ceiling show none. Crucially, points from different data budgets intermingle rather than stratifying—e.g. 5M at 100% data (utilization $\approx$96%, $+1.8$pp) falls in the same region as 5M at 10% ($\approx$94%, $+2.6$pp) and 5M at 30% ($\approx$95%, $+3.2$pp), despite a $10\times$ difference in absolute token count. The advantage is thus predicted by how much headroom remains, not by which data budget produced it; by this analysis, a larger model trained on proportionally insufficient data would be predicted to show similar gains.

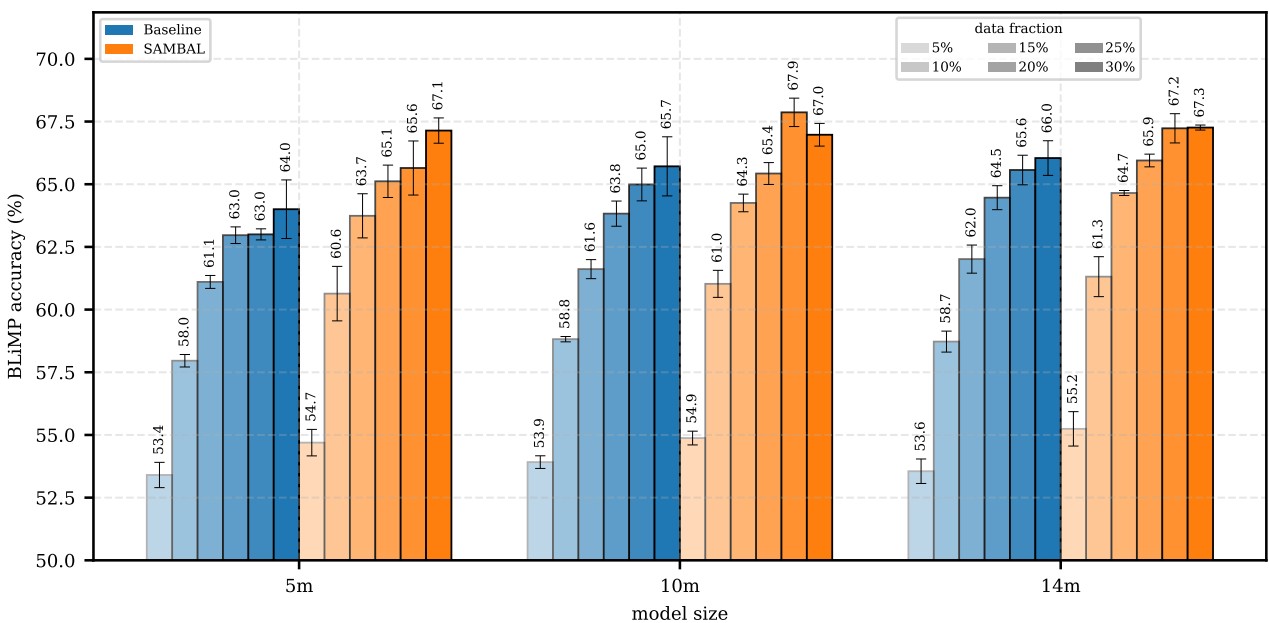

*Figure 5.* **Parameter and token efficiency** BLiMP vs model size and data fraction for 1M–3M tokens

## H. Lexical-regularization control

SAMBAL reduces lexical entropy as a side effect, so part of its BLiMP advantage could in principle stem from a simplified vocabulary rather than from suppressing semantics. (Removing SAMBAL's *syntactic* constraints instead hurts BLiMP; see the ablations of §5.) To isolate the lexical-entropy effect while retaining meaning, we train a control on unmodified original text filtered to sentences containing only the same top-25k vocabulary SAMBAL uses—preserving the original meaning but reducing lexical entropy—and match the 10% token budget of Fig. 4 (bottom). As Table 9 shows, the vocab-filtered control lands consistently above the baseline but below SAMBAL, so lower lexical entropy explains part but not all of SAMBAL's advantage.

*Table 9.* Lexical-regularization control (BLiMP %). A vocab-filtered baseline (original text restricted to SAMBAL's top-25k vocabulary) at the 10% token budget improves over the baseline but does not reach SAMBAL.

| Model size | baseline | Vocab-filtered | SAMBAL |
|---|---|---|---|
| 5M | 58.0 | $59.23 \pm 0.71$ | 60.6 |
| 14M | 58.7 | $60.08 \pm 0.29$ | 61.3 |
| 30M | 59.6 | $60.78 \pm 0.76$ | 61.6 |

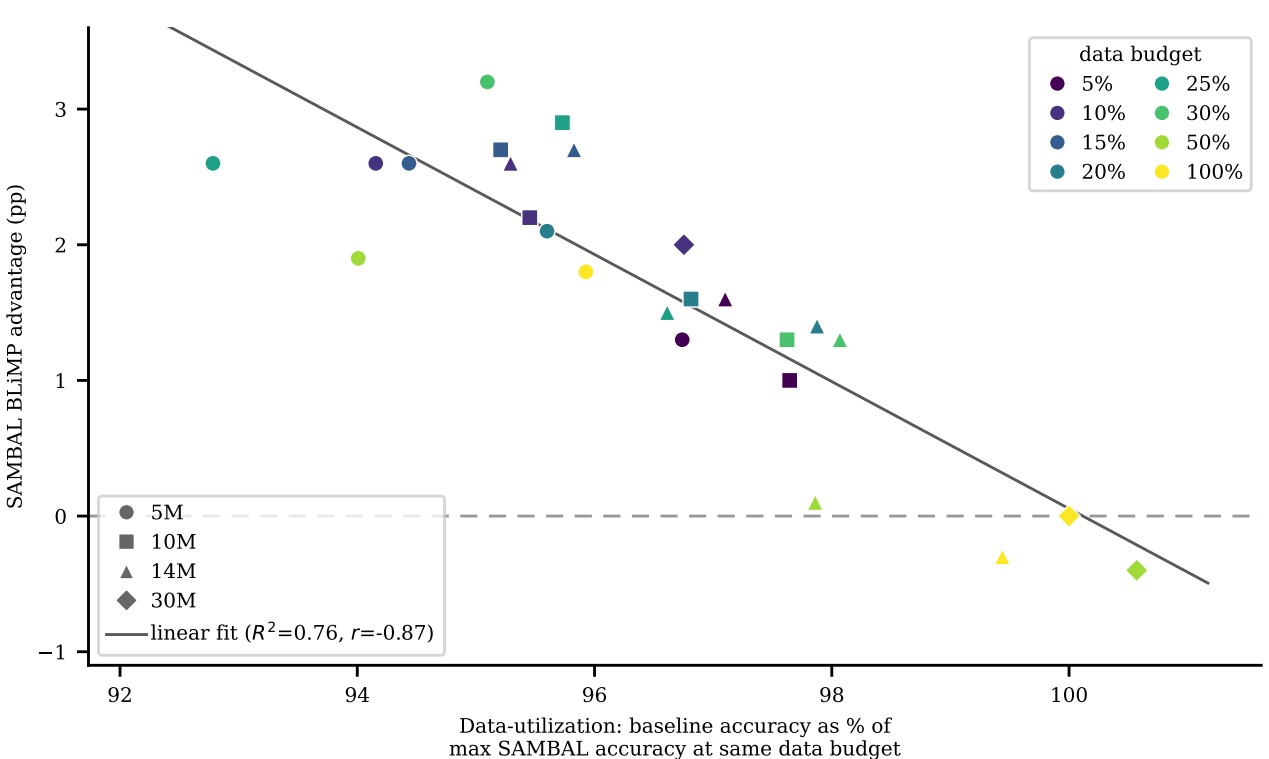

*Figure 6.* **The efficiency advantage tracks relative resource scarcity, not absolute scale.** SAMBAL's BLiMP advantage (pp) vs. data-utilization (baseline accuracy as a percentage of the maximum SAMBAL accuracy at the same data budget), over 25 configurations (8 data budgets × 4 model sizes). Color encodes data budget and marker shape encodes model size. The advantage is strongly predicted by data-utilization ($R^2 = 0.76$, $r = -0.87$); points from different data budgets intermingle rather than separating by absolute token count.

# I. Additional LoTR generations

| Prompt: | *'It's a dangerous business, Frodo, going out of your door' he used to say.* |
|---|---|
| SAMBAL | For Harrington, when I was little, there were three boxes of tubes on the guy, and they all came off as though they had sent some. And here I found 'Brazil,' this is what you know by Manipur Hungary. |
| | I'll be special with you.
In this formation!
We're not looking for any singing, glad?
Cats have traveled solely over the basis that they'd been threatening and marveling at one another.
Which is why we do it!
Give me less. |
| baseline | You're getting out, but you need to take care of him and get him home again.
Anyway, we should be hearing all about that.
No!
No!
I'm not so sure.
We can't do this without a frayed road.
- Sorry, Lushi. |
| | They wanted the man who saw him and didn't know how to handle it all. But they knew that there was no bad way for him to get away. So, they went back into his house to see what happened. One day, Minnie met a little fox named Frodo who would fly from one corner to another |
| SAMBAL finetuned | 'Then I hope you had better be off there.' He went on with his hand and drew himself up where the horse was clutching up Gollum along the slope. The two hobbits managed to go ahead, but they did not like it, even for a while; yet Frodo stared at them. 'Don't be absurd |
| | 'But you know more about it than I like, or this for certain.' He looked at Pippin and smiled aloud, but Sam could still see that the gleam in his eyes was not quite so easy as Frodo did; but it seemed and was becoming increasingly sick and at times so heavy that nothing else to do but follow him. |
| baseline finetuned | 'I've been doing nothing wrong. I don't like it! ' Gollum fell down the flask and clashed into the water. Frodo raised his head and took Frodo back to his chair. He was silent for a moment, staring at him with an eyebrows lit by the fire. 'Don't let me sit |
| | 'I am not going out at all,' said Sam. 'You know what I mean. Go on and get off with you. But you won't forget it.' The old man sat down on the ground again and looked at him. 'What do you mean? What are you trespassing about in your face, Frodo?' asked Gand |

# J. PubMed specialization

As a robustness check that rapid adaptation is not specific to literary text, we repeat the LoRA fine-tuning experiment of §4.6 on a set of PubMed abstracts matched to the LoTR corpus size ($\approx$2,600 abstracts). Table 10 shows the same trends as on LoTR: SAMBAL begins at much higher domain perplexity (255.4 vs. 40.5), yet after fine-tuning closes most of the gap (32.3) and substantially outperforms the best from-scratch model (58.54). Both models shed a few BLiMP points after specialization.

| Prompt: | *'The dark fire will not avail you, flame of Udyn. Go back to the Shadow! You cannot pass,' bellowed Gandalf.* |
|---|---|
| SAMBAL | "Political danger!" bent the part to mark his present hands and work on the shirt. "I am fed up with my own scientists to discover an open deal of mud. A father at once made their contribution towards a world in the metal of his ants, but for some death it was not sod for them to |
| | Where is it? Equally available? It relies on the young study of Oakdale which has been called 'effectiveness'. but is hired by Australian, ... and rural McGlory Jura itself, in a world that it comes like an object of school. The happy family of Englishman and Navaj |
| baseline | "But I do think that is all this is done." The other man happily scampered towards him as he looked at himself in his eyes. But eldest hut was partly going, and riding down the road with Joshua to be brought up with a gripping cloak. "Good evening, big boys. Oh, |
| | "You are going after me!" exclaimed Tilly, her voice barely above the workbench. "I am going after you; I can't bear to let your brother leave."
"But why? There will not be any time for a fray to go out until you start?"
"Well, you must hurry up so soon," said |
| SAMBAL finetuned | 'Then I must take your Ring.' he said. 'I am going to try and find Dain here beside him.' 'Yet you cannot! ' cried Boromir; but they had been too sure that he would not reach the Shire for a while; yet he could see a look in his heart again, as if he were looking |
| | 'But we must sleep. We will lay on the boughs of Tirithlode at night.' They drew up their cloaks and turned far northward towards the hills, but they were not much troubled; for their threads writhed in a sheer scored behind, and then shrill mists quaved. Far away |
| baseline finetuned | 'Then I have come into this place where there is no light! ' He bowed and stood up. Sam ran round him with his head drawn back. 'You are coming quick, Mr. Frodo,' he said. 'Wait for me!' He went on. Pippin sat up and looked out. The sky was black as the night had |
| | 'I am not eager for a flight or two, but for some long web and endless search in which I must first find the lost house.' 'Then what do we do?' said Sam at last. Before he could speak they went to the banner with Gimli tucked his pony carved out of stone; then their hands slanted |

# K. Additional discussion: terminology and line drawing for SAMBAL

## K.1. Syntax, semantics, grammar, and "world knowledge"

We use *syntax* to denote the formal constraints that govern well-formedness: category/feature compatibility, hierarchical dependencies, agreement, binding, and complement realization. We use *semantics* to denote meaning-bearing content: open-class lexical meanings, their compositional contribution, and the background knowledge that makes an utterance informative or plausible.

In the paper we frequently use *grammar* vs. *world knowledge* as a convenient shorthand. This shorthand is intentionally operational rather than precise: there exist meaning-like distinctions that are grammaticized and therefore directly affect acceptability. For SAMBAL, the practical question is whether a property must be preserved to avoid creating ungrammatical outputs (or flipping the label in a minimal pair), even if that property is not "pure syntax."

## K.2. Relation to the formal/functional competence distinction

Mahowald et al. (2024) distinguish *formal linguistic competence* from *functional linguistic competence* in evaluating LLMs. Their characterization of *formal* competence includes not only phonology/morphology and syntactic combination rules, but also *enough lexical meaning to know which words can go in which slots* and sensitivity to idiosyncratic constructions; *functional* competence involves using language in the world and draws on non-linguistic cognition (e.g., reasoning, world knowledge, situation tracking, social cognition). SAMBAL's goal is narrower and more adversarial: we aim to keep the morphosyntactic scaffold that supports acceptability, while explicitly suppressing both encyclopedic facts and many of the

*Table 10.* Specialization on PubMed with LoRA fine-tuning.

| Model | PubMed PPL | BLiMP Acc. |
|---|---|---|
| baseline (Long) | 40.5 | 79.2 |
| baseline (Long) fine-tuned | 23.2 | 72.1 |
| SAMBAL (Long) | 255.4 | 79.3 |
| SAMBAL (Long) fine-tuned | 32.3 | 71.0 |
| Best PubMed from-scratch model | 58.54 | 55.74 |

low-level plausibility cues that ordinarily make utterances coherent and interpretable.

### K.3. Why the line is hard to draw

The preserve/ablate boundary is inherently fuzzy because many semantic distinctions have grammatical reflexes:

- **Argument structure vs. plausibility:** verb meaning correlates with subcategorization. Preserving c-selection (e.g., whether a verb takes NP/PP/CP/to-inf) is grammatical; preserving "appropriate" thematic roles is plausibility.

- **Count/mass and determiner compatibility:** whether a noun behaves like a mass or count noun can determine whether a string with a given determiner is acceptable.

- **Animacy/humanness and binding/morphology:** pronoun/reflexive inventories encode properties that look semantic but are checked by the grammar.

- **Polarity sensitivity:** NPIs and the environments that license them are tightly linked to acceptability, even though they are also connected to meaning in the broad sense.

- **MWEs and constructional idiosyncrasy:** many MWEs behave like single lexical units; substituting within them often destroys grammatical well-formedness or yields parser-incompatible structures.

Because of these interactions, SAMBAL uses an explicitly operational criterion: if ablating a property reliably breaks well-formedness or flips acceptability in grammar-focused minimal pairs, we treat it as "grammar-relevant" for our purposes.

### K.4. Borderline cases: gender/humanness and polarity sensitivity

**Gender and humanness/animacy.**   Gender and humanness/animacy are borderline because they look like properties of referents, yet they are realized as morphosyntactic distinctions that directly affect acceptability (especially in pronominal/reflexive paradigms and binding). Ablating them would frequently create ungrammatical strings (or change which member of a minimal pair is acceptable), rather than merely reducing topical interpretability. We therefore preserve these features by default and, when needed, constrain replacements so that these values remain stable.

This choice is also evaluation-driven: BLiMP contains minimal pairs that directly hinge on reflexive morphology (e.g., proper-name antecedents selecting *herself* vs. *himself*), and gender is entangled with other syntactic phenomena being tested (e.g., quantified antecedents and binding configurations). Changing gender/humanness would confound these tests.

**NPIs and NPI licensors.**   Polarity sensitivity is similarly borderline: NPIs (e.g., *any*, *ever*) and the elements that license them (negation, downward-entailing quantifiers/determiners, and other polarity-conditioning material) can determine whether a sentence is acceptable. If SAMBAL were to freely relexicalize polarity items or their licensors, it could inadvertently flip grammaticality in NPI-focused minimal pairs and introduce "errors" that are not about syntactic well-formedness in the intended sense. Consequently, we preserve NPIs and NPI licensors as protected material in the default setting, treating them as grammar-critical guardrails even though they are meaning-adjacent.

