# OpenReview forum: "Learning syntax without semantics: Disentangled tiny language models"
_ICML.cc/2026/Conference — ICML 2026 regular_

### Official Review · Reviewer_ETH9 · 2026-02-22

**Soundness:** 2
**Presentation:** 3
**Significance:** 2
**Originality:** 2
**Overall Recommendation:** 3
**Confidence:** 4

**Summary:**

This paper introduces a data manipulation algorithm that aims at preserving syntactic structure while destroying semantic and world knowledge signals from a given corpus. A model trained on this corpus is shown to be strong in syntax, weak or at chance in world knowledge, and more efficient than the baseline model in terms of its syntax performance when the number of training tokens is kept constant.

**Compliance With Llm Reviewing Policy:**

Affirmed.

**Final Justification:**

While the authors' rebuttal was convincing in that dissociating syntax from semantics is of interest to the ML community, and I increased my score from 2 to 3, I lean towards rejection for the following reasons: (1) the notions of semantics and world knowledge are conflated, (2) although the efficient domain adaptation is attractive, its usefulness is very limited (only shown in Lord of the Ring corpus) and its practical implication remains unclear.

**Key Questions For Authors:**

1. The terms semantics and world knowledge seem to be used interchangeably in this paper. I am not particularly familiar with EWOK, but is model performing poorly at EWOK equivalent to model lacking semantic capabilities? Could it not be that the model understands semantics but lacks world knowledge? For a more targeted evlauation on semantics, why not use a more standard semantic evaluation, such as NLI?

2. For the corpus re-lexicalization, it relies on the initial parse of the corpus, as I understand it. How was the accuracy of the initial parse on deprel, pos tag, etc. which will propagate downstream? Also was there a manual check on the re-lexicalized corpus?

**Limitations:**

1. The paper introduces a way to train syntactically capable but semantically incapable model. This seems a bit trivial to me: modularizing these 2 domains within a model could be very valuable, but simply taking away one while keeping the other seems like training a model that's strictly worse (I will comment on subsequent FT and efficiency below).

2. Although the model seems to be more efficient in syntactic learning, the explanation is purely speculative, and not very informative. Also, the paper argues that domain-specific (semantic) knowledge can be injected via FT methods like LoRA. However, if semantic knowledge is to be taught later anyways, what was the point of only teaching syntax to begin with? In practice, what's the practical application of this method?

3. The model architecture is not mentioned (GPT-BERT, but not clear what that means because these are 2 very different models), and the parameter size is too small (30M) to judge its generalizability to different scales.

**Strengths And Weaknesses:**

Soundness:
- The paper is sound. Syntax-semantics distinction and evaluation seems a bit unclear, which I detail in the next section.

Presentation
- The paper was generally easy to understand and well presented.

Significance:
- While I enjoyed reading the paper, the significance of this paper is better appreciated by the (computational) linguistics and resources communities, rather than by an ML research community. The contribution mainly comes from the re-lexicalization algorithm and dataset creation, and it's not likely to impact the broader ML community.

Originality:
- The form-meaning separation is a long standing question in linguistics, and has been widely explored in computational linguistics using language models as the authors cite (e.g., different evaluations in BabyLM challenges). Yet, as far as I know, training a model entirely on semantically bleached sentences seems somewhat novel.

A few typos:
- LL137-140 right column: this dog runs vs this dog runs; weather
- L222 let column: that of ?

---

> ### Author Rebuttal · Authors · 2026-03-31
>
> Thank you for your review, which highlights several places where we need to clarify our presentation.
>
> For a discussion of scaling, please refer to our response (2. Scaling) to reviewer 8FNB.
>
> For a concrete piece of evidence getting at why SAMBAL may be more efficient, see our response (3. Lexical regularization control) to reviewer 8FNB.
>
> ---
> ### 1. Why teach syntax first if knowledge will be added later?
>
> Thank you for the question. This is something we will be sure to clarify in the revision. As you note, "modularizing these 2 domains within a model could be very valuable" — we agree, and that is the basis of our approach. SAMBAL pretraining encodes syntax in the base weights, and LoRA fine-tuning adds domain knowledge in a lightweight, modular adapter (only the added LoRA weights are trained).
>
> A practical application is a scenario where we need a linguistically-capable model that knows as little as possible beyond a small target domain, and the target domain is too small to train a model from scratch, as is the case in our Lord of the Rings demo (Sec. 4.6).
>
> ---
> ### 2. Semantics vs. world knowledge
>
> This is a very good point. We focused on defining what syntactic signal to preserve and treated everything else as "semantics" to be suppressed. Our use of that term was broad, encompassing everything from open-class lexical meaning to basic world knowledge. In the revised manuscript we will be more careful to clearly distinguish these. The operational target of SAMBAL is suppression of plausibility and world-knowledge cues while preserving grammar-critical signal. This is why some meaning-adjacent features (countability, humanness/gender, c-selection) are retained: they are cases where meaning has direct grammatical consequences.
>
> ---
> ### 3. NLI experiment
>
> We agree that poor EWoK performance should not be read as "no semantics." EWoK is our primary probe because it is directly scorable via LM probabilities without any additional classifier, and was designed around context-dependent plausibility judgments. It targets basic world-model behavior rather than encyclopedic factual recall.
>
> Per your suggestion, we ran SNLI as a complementary evaluation: the baseline reaches **67.4%** and SAMBAL reaches **60.1%**, while chance performance is 33.3%. We interpret this cautiously — the original SNLI paper reports 50.4% for a baseline without lexical features, so above-chance SNLI performance can be obtained from shallow structural cues alone rather than broad semantic competence.
>
> ---
> ### 4. ML relevance
>
> We respectfully disagree the paper would be better appreciated by the computational linguistics community. Our core question is whether capabilities entangled in standard LM pretraining can be separated through data design, and what this means for efficiency and modularity. To cite a few relevant ICML publications: Allen-Zhu & Li [1] intervene on pretraining data to study LM knowledge acquisition. The MOSS workshop at ICML 2025 [2] features work on disentangling statistical patterns from factual knowledge at small scale. Our knowledge-free, syntax-competent base that adapts rapidly via LoRA offers an alternative to post-hoc knowledge editing and unlearning such as [3]. Capability-specialized LMs [4] and controlled data interventions to isolate LM capabilities [5] are also established contributions at this venue.
>
> [1] Allen-Zhu & Li, "Physics of Language Models: Part 3.1, Knowledge Storage and Extraction", ICML 2024.
> [2] Methods and Opportunities at Small Scale (MOSS), ICML 2025 Workshop.
> [3] Liao et al., "AnyEdit: Edit Any Knowledge Encoded in Language Models", ICML 2025.
> [4] Fu et al., "Specializing Smaller Language Models towards Multi-Step Reasoning", ICML 2023 Oral.
> [5] Singh et al., "What needs to go right for an induction head?", ICML 2024 Spotlight.
>
>
> ---
> ### 5. Parser quality and manual check
>
> Good question. The parser/tagger used in SAMBAL is the spaCy `en_core_web_trf`, whose reported accuracies are 0.98 on POS, 0.94 LAS, and 0.95 UAS. We also manually spot-checked 100 sampled SAMBAL sentences and found 87% of them fully grammatical. So the transformed corpus is clearly imperfect, and we will make that more explicit. Ultimately, downstream performance on BLiMP and SyntaxGym is evidence that this level of fidelity is sufficient.
>
>
> ---
> ### 6. GPT-BERT model
>
> GPT-BERT (Charpentier & Samuel 2024) is a hybrid causal/masked language model that combines GPT-style autoregressive and BERT-style masked objectives. We chose it because it is SOTA on BLiMP at this scale. While the reference is provided the beginning of Sec. 4.1, we will be sure to describe this in more detail in the revised manuscript.

---

> > ### Author Rebuttal · Reviewer_ETH9 · 2026-04-01
> >
> > Thank you for the thorough response.
> > * 1, 4, 5, 6: Noted, thank you for the clarification.
> > * 2, 3: I think the use of these terms (semantics, world knowledge, etc.) was a bit unclear, but the author response made it clearer. Yet, I think the fact that the disentangled model still has an NLI score way above chance is non-trivial, and affects the core claim of this paper. It may be true that the surface cues are sufficient to scroe 60% on NLI, but readers have no way of knowing to what extent "syntax" and "semantics" are properly disentangled as the authors claim, if a non-trivial above-chance score on a semantic benchmark is dismissed as such.
> > * Given these, the authors did clarify many, but not all, of the concerns I have raised, hence I'm adjusting the score.

---

> > > ### Author Response · Authors · 2026-04-03
> > >
> > > Thank you for your reply.
> > >
> > > Regarding SNLI performance and retained semantic capabilities, we have an additional piece of evidence:
> > >
> > > SNLI consists of (premise, hypothesis) sentences pairs and the task is to classify if the hypothesis is entailed by the premise, contradicts it, or is neutral. [Gururangan et al. 2018](https://arxiv.org/abs/1803.02324) found that a classifier that _sees only the hypothesis_ reaches 67% accuracy, and denoted the remaining 33% of test examples SNLI-hard. On SNLI-hard, our baseline model obtains **44.3%** and SAMBAL obtains **37.5%**. This is much closer to chance level, while retaining the ~7% gap vs baseline seen on the full test set.
> > >
> > > Also, this still doesn't control for the known surface-level shortcuts that involve both premise and hypothesis, for example word-overlap count. Those alone achieve 50.4% accuracy, as reported by [Bowman et al. 2015](https://arxiv.org/abs/1508.05326).

---

### Official Review · Reviewer_8FNB · 2026-03-12

**Soundness:** 3
**Presentation:** 3
**Significance:** 3
**Originality:** 3
**Overall Recommendation:** 4
**Confidence:** 4

**Summary:**

The paper introduces SAMBAL, a constrained recoding pipeline that turns natural text into a syntactically well-formed but semantically anomalous corpus by replacing open-class words while keeping morphosyntax, agreement, and the rough complement structure intact. The authors use this corpus to pretrain small language models and test whether a substantial portion of syntactic competence can still be learned when signals of semantic plausibility and world knowledge are heavily weakened.

Experimentally, the authors show that models trained on SAMBAL retain comparable performance on syntactic benchmarks (especially BLiMP and parts of SyntaxGym), while losing the expected behavior on world-knowledge tests. They also show a stronger preference for grammaticality when grammar and plausibility come into conflict, exhibit more “syntax-oriented” representations, and may be more data- and parameter-efficient in low-resource settings.

**Compliance With Llm Reviewing Policy:**

Affirmed.

**Final Justification:**

After reading the rebuttal, I think the authors addressed my main concerns sufficiently. The additional lexical-regularization control was especially helpful, and the clarification about the grammar-plausibility benchmark makes the claims more convincing. I still see the paper mainly as a small-scale, low-resource study, so the final version should make that scope clear and avoid stronger claims about a complete syntax/semantics separation. With this narrower framing, I keep my overall recommendation at Weak Accept.

**Key Questions For Authors:**

- Does this mean that SAMBAL is basically a curriculum-style trick that really matters only under severe under-resourcing, rather than reflecting some fundamental property of syntax/semantics separation?

- Could you run a control condition: a conditional baseline trained on text with a reduced vocabulary (say, the top-K lemmas by frequency), but without SAMBAL’s syntactic constraints? If that model shows a comparable gain on BLiMP, then the main effect may not be semantic suppression at all, but just lexical regularization.

- How much of the result is actually explained by lower lexical entropy / a simplified vocabulary, rather than by suppressing semantic signals?

**Limitations:**

yes

**Strengths And Weaknesses:**

Strengths:
1. The paper asks a clear and important empirical question: whether one can substantially weaken semantic / knowledge-based signals in the training data while still preserving the ability to learn syntax.
2. The experimental evaluation is broad and fairly well-rounded: it includes standard syntactic benchmarks, world-knowledge tests, carefully constructed minimal pairs, representation analyses, scaling across data and model size, and domain adaptation.
3. SAMBAL is presented as a fairly careful, linguistically motivated pipeline, rather than just a crude or ad hoc augmentation trick. That makes the overall setup more convincing.
4. The paper does a good job of combining benchmark-level results with more diagnostic analyses, which makes the story stronger.

Weaknesses:

1. The main claim about “separating syntax from semantics” feels stronger than what the experiments really establish. The paper shows a substantial suppression of some lexical-semantic and world-knowledge signals, rather than a clean syntax/semantics separation.

2. SAMBAL still preserves many meaning-relevant cues, including function words, morphosyntax, countability, humanness, gender, subcategorization and context-conditioned sampling. Because of that, it is not fully clear that this is really “syntax without semantics,” as opposed to “syntax plus a preserved layer of meaning-adjacent information.”

3. Some of the most striking conclusions rely on custom benchmarks, especially the grammar-vs.-plausibility conflict benchmark. But that benchmark is at least partly conditioned on baseline behavior (e.g., selecting cases where the baseline already prefers plausibility)

4. The efficiency gains in terms of data and parameter scaling seem most visible mainly for smaller models and low-resource setting, and they mostly fade as the budget grows. In addition, the method depends quite heavily on high-quality English-specific tools and resources, which raises some concern about generality and about how much of the syntactic competence is effectively injected through the pipeline rather than learned from data in a broader sense.

---

> ### Author Rebuttal · Authors · 2026-03-31
>
> Thank you for your review.
>
>
> For a discussion of "syntax without semantics," please see our response (1. Semantic leakage channels) to reviewer zhJN.
>
> ---
> ### 1. Baseline-conditioned benchmark
>
> Yes, the grammar-vs.-plausibility conflict benchmark is baseline-conditioned by construction, so it should be read as an adversarial stress test of grammar/plausibility competition, not as a general syntax score. We will make that caveat more prominent. Our main evidence does not depend on this benchmark alone; in addition to the other benchmarks, we now have a complementary SNLI result described in our response (3. NLI experiment) to reviewer ETH9. We would also argue that the results of the grammar plausibility swap probe, shown in Figure 1, are equally striking and do not involve baseline-conditioning.
>
> ---
> ### 2. Scaling
>
> We acknowledge that our experiments do not establish whether SAMBAL's efficiency gains persist at larger absolute scales, and we leave direct verification to future work. That said: 1) the practical value of modular decomposition — pretraining a syntax-competent base and injecting domain knowledge post-hoc — does not depend on efficiency gains scaling, 2) human-scale language modeling is an active area of study in its own right (BabyLM Challenge, MOSS workshop at ICML), and 3) the scientific contributions of the paper — demonstrating that syntactic competence survives aggressive semantic ablation, characterizing the syntax/semantics boundary, and showing qualitatively different behavior on conflict probes — are not contingent on scale.
>
> On the empirical question: There is an important distinction between _small absolute scale_ and _relative resource scarcity_. We believe the efficiency benefit is not confined to small models or small data in absolute terms. To test whether SAMBAL's advantage tracks the latter, we re-analyzed the existing scaling sweep, plotting SAMBAL's BLiMP advantage (pp) against a data-utilization measure: baseline accuracy as a percentage of the maximum SAMBAL accuracy at the same data budget. This measures how far the baseline is from the best performance achievable at a given data budget. Across 25 configurations spanning 8 data budgets (5%–100%) and 4 model sizes (5M–30M), the relationship is strongly linear ($R^2$ = 0.76, r = −0.87) with a negative slope of about -0.5pp per percentage point of utilization. Crucially, points from different data budgets intermingle rather than stratifying, despite a 10x difference in absolute token count. _The advantage appears whenever the model is parameter-constrained relative to its data budget, and vanishes once the model has enough capacity to fully exploit the available data._ We will include a plot of this analysis in the revised manuscript.
>
> ---
> ### 3. Lexical regularization control
>
> Whether SAMBAL's performance attributes to suppressed semantics or lexical regularization alone is a very good question. A control run without _syntactic constraints_ would hurt BLiMP performance, as seen from our ablations. However, we test this as follows: we train a control on unmodified original text after filtering to those sentences containing only the top-25k vocabulary, the same vocabulary SAMBAL uses. So this dataset retains the original meaning, but has reduced lexical entropy. We match the 10% token budget used in Fig. 4 (bottom).
>
> | Model size | Baseline | Vocab-filtered | SAMBAL |
> |---|---|---|---|
> | 5M | 58.0 | 59.23 +/- 0.71 | 60.6 |
> | 14M | 58.7 | 60.08 +/- 0.29 | 61.3 |
> | 30M | 59.6 | 60.78 +/- 0.76 | 61.6 |
>
> The resulting BLiMP scores are consistently above the baseline but still below SAMBAL. So lower lexical entropy / lexical regularization may explain part but not all of SAMBAL's advantage.
>
> We also note that lexical regularization alone cannot deliver SAMBAL's other contributions: a vocab-filtered model still encodes world knowledge and plausibility preferences, lacking the modularity that enables rapid domain adaptation (§4.6). Moreover, as we discuss in §5, aggressive vocabulary reduction trades off against downstream coverage — restricting replacements to the BLiMP vocabulary improves BLiMP by ~2pp but degrades SyntaxGym and increases adaptation perplexity.

---

> > ### Author Rebuttal · Reviewer_8FNB · 2026-04-03
> >
> > Thank you for the detailed and helpful rebuttal. I found the additional analyses useful, especially the lexical-regularization control, the clarification that the grammar–plausibility conflict benchmark should be interpreted as an adversarial stress test rather than a general syntax benchmark, and the more careful framing of the paper’s claims around partial disentanglement / suppression rather than a perfectly clean syntax–semantics separation.
> >
> > I still think the paper is strongest as a small-scale, low-resource study, and that this scope should be made explicit in the final version. With that framing, however, my main concerns have been adequately addressed. I am therefore keeping my overall recommendation at Weak Accept.

---

### Official Review · Reviewer_zhJN · 2026-03-13

**Soundness:** 3
**Presentation:** 2
**Significance:** 2
**Originality:** 3
**Overall Recommendation:** 4
**Confidence:** 3

**Summary:**

This paper studies whether tiny language models can acquire syntactic competence while suppressing semantic plausibility and world-knowledge cues. The authors introduce SAMBAL, a constrained relexicalization pipeline that transforms natural text into grammatical nonsense while preserving morphosyntax and coarse argument structure, then pretrain small GPT-BERT models on this transformed corpus. Empirically, the paper reports near-parity with standard pretraining on BLiMP and most SyntaxGym suites, near-chance performance on world-knowledge and plausibility probes, improved low-resource BLiMP scaling, and rapid LoRA adaptation to a small domain corpus. The paper also includes custom conflict probes and a representation analysis intended to show reduced syntax-semantics entanglement.

**Compliance With Llm Reviewing Policy:**

Affirmed.

**Final Justification:**

The rebuttal has addressed my concerns. However,  upon checking the other review comments (especially Reviewer ETH9), I decide to keep my rating.

**Key Questions For Authors:**

See the weaknesses above.

**Limitations:**

No. Please refer to the weaknesses above.

**Strengths And Weaknesses:**

**Strengths:**
- The paper targets a sharp and interesting question: can syntax be learned with semantics and world knowledge deliberately suppressed, rather than merely emerging jointly during standard pretraining?
SAMBAL is a fairly concrete and carefully engineered intervention. Table 1 and Algorithm 1 make the preserve-vs-ablate design reasonably explicit, and the operational distinction between c-selection and s-selection is useful.
- The empirical evaluation is broader than a single benchmark. The paper includes standard syntax benchmarks, a world-knowledge benchmark, custom conflict probes, scaling experiments, and a small adaptation study.
- SAMBAL points cluster near zero on semantic-plausibility margin while remaining strongly positive on syntax margin, whereas the normal LM shows the opposite tendency. Even though the probe is custom, the visualization clearly communicates what the authors are trying to isolate.
- The main benefit of SAMBAL is not universal improvement, but specifically better performance in very small data/model regimes. That is a more believable and more interesting claim than promising gains everywhere.

**Weaknesses:**
- This work attempts to investigate the problem of syntax-semantics disentanglement exclusively at 5M–30M parameters with ≤10M tokens. The authors provide no theoretical or empirical evidence that the approach scales to practical model sizes (1B+ parameters, 100B+ tokens). The efficiency gains may simply reflect that SAMBAL learns a strictly easier task, and it remains unclear whether disentanglement persists when models have sufficient capacity to jointly learn syntax and semantics. Without scaling experiments, the practical relevance is severely limited.

- This paper's key contribution comprises a method to ablate semantics while preserving syntax, yet multiple leakage channels remain: function word distributions, constructional patterns, corpus-conditioned sampling (which uses semantic co-occurrence statistics), and preserved features like gender/humanness. The ablation showing that removing context-conditioned sampling drops BLiMP by 4.8 points suggests some syntactic success depends on distributional semantic cues. The paper lacks adversarial evaluation to characterize when these leakage channels compromise disentanglement.

- SAMBAL collapses to near-zero accuracy on SyntaxGym reflexive suites, a core syntactic phenomenon. The explanation that 10M tokens is insufficient for learning reflexive binding under interference directly contradicts the claim that syntax can be efficiently learned from limited data. If the approach fails on basic long-distance dependencies, its utility for building syntactically competent models is questionable.

---

> ### Author Rebuttal · Authors · 2026-03-31
>
> Thank you for your review.
>
> For a discussion of scaling, please refer to our response (2. Scaling) to reviewer 8FNB.
>
> ---
> ### 1. Semantic leakage channels
>
> We appreciate this point and apologize if "syntax without semantics" in the title suggests a cleaner separation than we intend to claim. As we discuss in Section 3.1, Section 5, and Appendix H, the preserve/ablate boundary is inherently fuzzy because many semantic distinctions have grammatical consequences — e.g. gender, humanness, and countability all affect well-formedness directly. Table 1 identifies these borderline cases explicitly, and Appendix H.3 is devoted to "why the line is hard to draw." We will revise the language wherever possible to more consistently highlight these caveats.
>
> Reviewer 8FNB's characterization as "syntax plus a preserved layer of meaning-adjacent information" is quite fair. We view the precise discussion of this fuzzy boundary — what must be preserved, what can be ablated, and why — as itself a contribution of the paper. Our goal is to push the separation as far possible and characterize the residual entanglement, not to claim it is zero.
>
> **Context-conditioned sampling:** We can investigate this by further evaluation of the ablated model without context-conditioned sampling. Because the standard SAMBAL model with context-conditioned sampling already performs near chance on EWoK and shows minimal preference for semantic plausibility on our custom probes, we would not expect even less semantic understanding from the ablated model on these probes. However, as we detail in our response (3. NLI experiment) to reviewer ETH9, we have conducted additional semantic evaluation on the SNLI benchmark. While the baseline obtained **67.4%** and standard SAMBAL **60.1%**, the ablated model got **59.7%**, roughly equivalent. This can be taken as one piece of evidence that context-conditioned sampling doesn't inject significant semantic cues.
>
> ---
> ### 2. SyntaxGym reflexive performance
>
> We appreciate this concern, and have continued investigating the cause of the failure on the reflexive suites. After re-checking, we found that the relevant _singular-controller + plural-distractor + reflexive_ pattern and its reverse, which the SyntaxGym reflexive suites test, are actually _entirely absent_ from the training corpus. Our earlier "<50" count was an error, referring to a broader family of reflexive cue-conflict cases that do not teach the relevant rule.
>
> To test whether the collapse reflects a structural limitation or a missing-pattern/data-sparsity problem, we augmented the training corpus with relevant examples: we added two SAMBAL-ablated sentences created from each of the 228 grammatical SyntaxGym reflexive items (and — because reflexive-only augmentation slightly hurt number-suite behavior — we added examples from the number-suite items as well). This increased training corpus size by about 6000 tokens. Training short-regime models on the augmented SAMBAL training corpus raises the reflexive-suite averages to  **98.0%**, with overall SyntaxGym average improving to **66.8 +/- 6.9**. Augmenting the baseline training corpus with the same examples yielded **100%** on reflexive suites and **73.8 +/- 1.3** on SyntaxGym overall. We will update the manuscript with these findings.
>
> We therefore interpret the original failure as a genuine weak point of the 10M-token regime, but one that appears to be driven by missing pattern coverage, not by an inability of SAMBAL-trained models to learn long-distance binding in principle. We would argue that this does not "contradict the claim that syntax can be efficiently learned from limited data", but rather reflects that this particular syntactic pattern is adversarial and quite uncommon in natural corpora. As discussed in the paper, we believe that the 10M token corpus is "near-threshold" for learning this behavior. The improvement with the addition of limited examples lends weight to this interpretation.

---

### Official Review · Reviewer_WD9X · 2026-03-13

**Soundness:** 4
**Presentation:** 3
**Significance:** 4
**Originality:** 4
**Overall Recommendation:** 5
**Confidence:** 3

**Summary:**

The authors present a study about how well language models can learn syntax while
suppressing semantic plausibility and world knowledge. They propose a novel pipeline,
SAMBAL, that transforms natural text into grammatical nonsense through constraint
re-lexicalization. They consider tiny LMs (5M parameters) trained with altered corpora and
compare them with a baseline model (30 M parameters) trained on the original corpus from the
BabyLM competition. Their findings suggest that SAMBAL-based pre-training matches standard
pre-training on grammatical benchmarks such as BLIMP and SyntaxGym datasets. Additionally,
they also show that SAMBAL-based models adapt rapidly to an in-domain corpus LOTR, where
competent models are infeasible when trained from scratch.

**Compliance With Llm Reviewing Policy:**

Affirmed.

**Final Justification:**

While the authors have addressed my concern regarding extending the applicability to domain-specific datasets, I would like to stick with my overall recommendation of the paper.

**Key Questions For Authors:**

Typo: L222, left column, fix the “?”

**Limitations:**

yes

**Strengths And Weaknesses:**

Strengths
- The paper is well written and easy to follow.
- The idea of learning syntax while suppressing semantic and world-knowledge cues is
novel and well motivated.
- The SAMBAL pipeline is a strong contribution that may further encourage research in
this direction.
- The finding related to efficiency regarding sample and parameter gain in lower resource
regimes is inspiring and can be applied to domain-specific settings.

Weaknesses
- The authors’ claim about SAMBAL models to adapt to in-domain corpora is important yet
not generalizable. It would be interesting to conduct similar experiments with a bit-far
domain related to world knowledge, such as any medical corpora (eg, PubMed) or
commonsense-related data

---

> ### Author Rebuttal · Authors · 2026-03-31
>
> Thank you for your review. We agree that the Lord of the Rings domain-adaptation study is only a small proof-of-concept, and see the full development of practical, modular domain specialization as a promising future direction. However, to demonstrate adaptation to a more technical domain here, we have followed your helpful suggestion. We ran the same experiment on a set of PubMed abstracts matching the size of the LoTR corpus, amounting to about 2600 training abstracts. We observed the same trends as on LoTR:
>
> |                          | PubMed test PPL | BLiMP Acc. |
> |--------------------------|-----------------|------------|
> | baseline (long)          |            40.5 |       79.2 |
> | baseline (long) ft       |            23.2 |       72.1 |
> | SAMBAL (long)            |           255.4 |       79.3 |
> | SAMBAL (long) ft         |            32.3 |         71 |
> | Best PubMed from-scratch |           58.54 |      55.74 |
>
> We will include this in the revised manuscript.

---

> > ### Author Rebuttal · Reviewer_WD9X · 2026-04-02
> >
> > Thanks to the authors for addressing my comments. In particular, for providing an additional study on PubMed data showing extension to other domains.
> >
> > While the overall trends appear similar, could you explain the score for SAMBAL(long)+Pubmed test (PPL)?
> >
> > While my raised concerns are somewhat addressed, other reviews still suggest further areas for improvement. Therefore, I would like to keep my overall scores for now.

---

> > > ### Author Response · Authors · 2026-04-03
> > >
> > > Thank you for your reply.
> > >
> > > The high PubMed test PPL (255.4) for the non-fine-tuned SAMBAL model is expected and in fact desirable: it indicates that SAMBAL successfully avoided learning domain-specific plausibility during pretraining. Importantly, despite starting from 255.4, SAMBAL closes most of the gap after LoRA fine-tuning (32.3), substantially outperforming the best from-scratch model (58.54). This demonstrates that just the core syntactic competence of the SAMBAL model provides a useful initialization for rapid domain adaptation.

---

### Decision · Program_Chairs · 2026-04-30

**Decision:**

Accept (regular)

**Comment:**

The paper puts forward SAMBAL, for moving from natural text into syntactically valid but semantically degraded data to study whether language models can learn syntax independently of semantics. The idea novel and well-motivated.The evaluation is broad, and shows that syntactic competence can be retained while removing semantics, especially in low-resource settings. This has promising implications for improving efficiency and supporting low-resource languages. Overall, the approach and results are insightful.